# Continuous Contrastive Learning for Long-Tailed Semi-Supervised Recognition

**Zi-Hao Zhou**[1,2*]    **Siyuan Fang**[1,2*]    **Zi-Jing Zhou**[3]    **Tong Wei**[1,2†]
**Yuanyu Wan**[4]    **Min-Ling Zhang**[1,2]

[1]School of Computer Science and Engineering, Southeast University, Nanjing, China
[2]Key Laboratory of Computer Network and Information Integration (Southeast University),
Ministry of Education, China
[3]Xiaomi Inc., China
[4]School of Software Technology, Zhejiang University, Ningbo, China
`{zhouzih, syfang, weit}@seu.edu.cn`

## Abstract

Long-tailed semi-supervised learning poses a significant challenge in training models with limited labeled data exhibiting a long-tailed label distribution. Current state-of-the-art LTSSL approaches heavily rely on high-quality pseudo-labels for large-scale unlabeled data. However, these methods often neglect the impact of representations learned by the neural network and struggle with real-world unlabeled data, which typically follows a different distribution than labeled data. This paper introduces a novel probabilistic framework that unifies various recent proposals in long-tail learning. Our framework derives the class-balanced contrastive loss through Gaussian kernel density estimation. We introduce a continuous contrastive learning method, CCL, extending our framework to unlabeled data using *reliable* and *smoothed* pseudo-labels. By progressively estimating the underlying label distribution and optimizing its alignment with model predictions, we tackle the diverse distribution of unlabeled data in real-world scenarios. Extensive experiments across multiple datasets with varying unlabeled data distributions demonstrate that CCL consistently outperforms prior state-of-the-art methods, achieving over 4% improvement on the ImageNet-127 dataset. Our source code is available at `https://github.com/zhouzihao11/CCL`.

## 1   Introduction

Semi-supervised learning (SSL) serves as a powerful approach for improving the generalization capabilities of deep neural networks (DNNs) in scenarios where labeled data is scarce [37, 59, 6, 23]. The core concept of SSL methods typically involves assigning pseudo-labels to unlabeled data and utilizing those with high confidence for model training [56, 71, 10]. However, many existing SSL algorithms presuppose a balanced label distribution across both labeled and unlabeled datasets. In real-world applications, datasets commonly exhibit a long-tailed label distribution [65, 28, 50, 67, 55], leading to biased pseudo-label generation favoring majority classes [40, 3, 68, 24]. This discrepancy challenges the effectiveness of SSL algorithms in addressing real-world datasets.

The exploration of long-tailed semi-supervised learning (LTSSL) has gained momentum to address the challenge of biased pseudo-label distribution arising from class imbalance in labeled and unlabeled data. Recent LTSSL approaches propose compensating for the learning of minority classes by

---

[*]equal contribution

[†]corresponding author

38th Conference on Neural Information Processing Systems (NeurIPS 2024).

distribution alignment [33, 63], data rebalancing [22, 38], and logit adjustment [64, 43] to rectify the pseudo-label distribution. However, existing approaches often assume the equivalence of the unlabeled data distribution with the labeled data or rely on predefined anchor distributions to estimate the unlabeled data distribution [64, 43]. Furthermore, these methods primarily focus on correcting model outputs without delving into the role of representation learning in improving performance.

This paper explicitly introduces an approach to obtain effective representations for long-tail learning by adopting an information-theoretic view of DNNs. We present a probabilistic framework that utilizes the deep variational information bottleneck method [1] to learn good representations and demonstrate its unification of recent long-tail learning proposals, such as logit adjustment [44] and balanced softmax [51], through approximating the density of class-conditional distribution in different ways. Specifically, our framework encompasses class-balanced supervised contrastive learning [73, 15] via Gaussian kernel density estimation. We extend this framework to address LTSSL by adapting the supervised contrastive loss to unlabeled data using "*continuous pseudo-labels*", derived from model predictions and propagated labels, to mitigate confirmation bias. To account for varying label distribution of unlabeled data, we progressively estimate the label distribution through a moving average and adjust model predictions to align with the estimated distribution.

In summary, our contributions are as follows:

1. We propose a probabilistic framework which unifies many recent proposals in long-tail learning. Specifically, popular class-balanced contrastive learning methods can be seen as special cases of our framework when approximating the density using a Gaussian kernel.

2. We generalize the proposed framework to LTSSL and present a continuous contrastive learning method based on reliable and smoothed pseudo-labels to address confirmation bias and improve the quality of learned representations.

3. We conduct extensive experiments across several LTSSL datasets with diverse label distributions of unlabeled data. The results show that our proposal substantially outperforms previous state-of-the-art methods.

## 2 A Probabilistic Framework for Long-Tail Learning

In this section, we first introduce a general framework for learning good representations. Then, we expand this framework to long-tail learning and illustrate how recent proposals can be regarded as specific instances of our framework through three ways for density approximation.

**Problem setup of long-tail learning.** We consider a $C$-class classification problem with instance space $\mathcal{X}$ and target space $\mathcal{Y} = \{1, \ldots, C\}$. Let $P_s$ and $P_t$ denote the source (training) and test distributions on $(\mathcal{X}, \mathcal{Y})$, respectively. We denote by $\mathbb{P}_s$ and $\mathbb{P}_t$ the corresponding probability density (or mass) functions. Given a training dataset $\{(\boldsymbol{x}_i, y_i)\}_{i=1}^{N}$, where $\boldsymbol{x}_i \in \mathbb{R}^d$ is the training sample and $y_i$ is the ground-truth label.

### 2.1 Learning good representations from information theoretical view

In this subsection, we rethink one of the most popular approaches to deal with representation learning, i.e., contrastive learning. We derive many recent proposals in this branch from an information-theoretic view. Let $Z$ denote the latent representation of $X$ induced by the encoder $\text{enc}(\cdot)$ parameterized by $\boldsymbol{\Theta}$. From an information-theoretic view, an optimal representation $Z$ is maximally informative about the target $Y$, and minimally "memorizes" $X$. The information bottleneck [60] adopts mutual information $I(\cdot)$ to measure information between two variables. Thus, optimal $Z$ can be obtained by maximizing the following objective:

$$\boldsymbol{\Theta}^* = \arg\max_{\boldsymbol{\Theta}} I(Z, Y; \boldsymbol{\Theta}) - \delta I(Z, X; \boldsymbol{\Theta}), \tag{1}$$

where $\delta \geq 0$ is a tradeoff parameter. The variational information bottleneck [1] solves the above objective by variational inference. Based on the definition of $I(\cdot)$, we can rewrite Eq. (1) as:

$$I(Z, Y) - \delta I(Z, X) = \int dy d\boldsymbol{z} \, \mathbb{P}(y, \boldsymbol{z}) \log \frac{\mathbb{P}(y \mid \boldsymbol{z})}{\mathbb{P}(y)} - \delta \int d\boldsymbol{z} d\boldsymbol{x} \, \mathbb{P}(\boldsymbol{x}, \boldsymbol{z}) \log \frac{\mathbb{P}(\boldsymbol{z} \mid \boldsymbol{x})}{\mathbb{P}(\boldsymbol{z})}, \tag{2}$$

where we omit $\boldsymbol{\Theta}$ for simplicity. Since $\mathbb{P}(y \mid \boldsymbol{z}) = \int d\boldsymbol{x} \, \mathbb{P}(\boldsymbol{x} \mid \boldsymbol{z}) \mathbb{P}(y \mid \boldsymbol{x})$ is intractable in Eq. (2), let $\widehat{\mathbb{P}}(y \mid \boldsymbol{z})$ be a variational approximation to $\mathbb{P}(y \mid \boldsymbol{z})$ and considering that the Kullback-Leibler (KL)

divergence $\mathrm{KL}[\mathbb{P}(y \mid z), \widehat{\mathbb{P}}(y \mid z)]$ is always positive, we have RHS of Eq. (2)'s lower bounded:

$$\int d\boldsymbol{x} dy d\boldsymbol{z} \mathbb{P}(\boldsymbol{x}) \mathbb{P}(y \mid \boldsymbol{x}) \mathbb{P}(\boldsymbol{z} \mid \boldsymbol{x}) \log \widehat{\mathbb{P}}(y \mid \boldsymbol{z}) - \delta \int d\boldsymbol{x} d\boldsymbol{z} \mathbb{P}(\boldsymbol{x}) \mathbb{P}(\boldsymbol{z} \mid \boldsymbol{x}) \log \frac{\mathbb{P}(\boldsymbol{z} \mid \boldsymbol{x})}{\mathbb{P}(\boldsymbol{z})}. \quad (3)$$

Suppose we use an encoder of the form $\mathbb{P}(\boldsymbol{z} \mid \boldsymbol{x}) \sim \mathcal{N}(\mathrm{enc}(\boldsymbol{x}), \varepsilon^2 \boldsymbol{I})$ and $\mathbb{P}(\boldsymbol{z}) \sim \mathcal{N}(\mathbf{0}, \boldsymbol{I})$, the second term of Eq. (3) equals the KL divergence $\mathrm{KL}[\mathbb{P}(\boldsymbol{z} \mid \boldsymbol{x}), \mathbb{P}(\boldsymbol{z})]$. Since $\mathbb{P}(\boldsymbol{z} \mid \boldsymbol{x})$ and $\mathbb{P}(\boldsymbol{z})$ are normal distributions, it can be rewritten as: $-\delta l(\frac{1}{2} \varepsilon^2 - 1 - 2 \log \varepsilon) - \delta \|\mathrm{enc}(\boldsymbol{x})\|^2$, where $l$ is the dimension of $\boldsymbol{z}$. For a deterministic model, $\boldsymbol{z}$ is almost unique for each $\boldsymbol{x}$, thus assuming $\varepsilon$ is a small constant close to 0. By integrating out $\mathbb{P}(\boldsymbol{z} \mid \boldsymbol{x})$ and discarding constant terms, maximizing Eq. (3) can be approximated by minimizing:

$$-\int d\boldsymbol{x} \mathbb{P}(\boldsymbol{x}) \int dy \mathbb{P}(y \mid \boldsymbol{x}) \log \widehat{\mathbb{P}}(y \mid \mathrm{enc}(\boldsymbol{x})) + \delta \|\mathrm{enc}(\boldsymbol{x})\|^2. \quad (4)$$

In the following of this paper, we denote the output of $\mathrm{enc}(\boldsymbol{x})$ as $\boldsymbol{z}$ (or $\boldsymbol{z}_{\boldsymbol{x}}$ for a particular sample $\boldsymbol{x}$) for simplicity. Minimizing Eq. (4) is equivalent to minimizing the following objective in the distribution of test data on each $\boldsymbol{x}$:

$$-\sum_{k \in [C]} \mathbb{P}_t(Y = k \mid \boldsymbol{x}) \log \widehat{\mathbb{P}}_t(Y = k \mid \boldsymbol{z}) + \delta \|\boldsymbol{z}\|^2. \quad (5)$$

Notably, Eq. (5) can be seen as a general framework for learning good representations. If $\mathbb{P}_s(Y) = \mathbb{P}_t(Y)$, $\mathbb{P}_t(Y = y \mid \boldsymbol{x})$ can simply be substituted by the ground-truth labels of training samples. However, in long-tail learning, the class-probability function $\mathbb{P}_t(Y = y \mid \boldsymbol{x})$ is different from that of the training data due to label distribution shift.

## 2.2 Probabilistic framework for long-tailed supervised learning

Since $\mathbb{P}_s(Y = y \mid \boldsymbol{x}) \neq \mathbb{P}_t(Y = y \mid \boldsymbol{x})$, we cannot directly solve Eq. (5). However, since long-tail learning typically assumes that $\mathbb{P}_t(Y)$ is uniform and we work with the label shift assumption, i.e., $\mathbb{P}_s(\boldsymbol{x} \mid Y = y) = \mathbb{P}_t(\boldsymbol{x} \mid Y = y)$, we can obtain $\mathbb{P}_t(Y = y \mid \boldsymbol{x})$ by Bayes' theorem:

$$\mathbb{P}_t(Y = y \mid \boldsymbol{x}) = \frac{\mathbb{P}(\boldsymbol{x} \mid Y = y)}{\sum_{k \in [C]} \mathbb{P}(\boldsymbol{x} \mid Y = k)} = \frac{\frac{1}{\mathbb{P}_s(Y = y)} \mathbb{P}_s(Y = y \mid \boldsymbol{x})}{\sum_{k \in [C]} \frac{1}{\mathbb{P}_s(Y = k)} \mathbb{P}_s(Y = k \mid \boldsymbol{x})}. \quad (6)$$

Throughout the paper, we use the notation $\mathbb{P}(\boldsymbol{x} \mid Y = y)$ to represent either $\mathbb{P}_s(\boldsymbol{x} \mid Y = y)$ or $\mathbb{P}_t(\boldsymbol{x} \mid Y = y)$. In practice, $\|\boldsymbol{z}\|^2$ can be omitted in optimization because normalization is commonly adopted in deep learning. In long-tail learning, minimizing Eq. (5) equals to minimizing:

$$-\sum_{k \in [C]} \frac{1}{\mathbb{P}_s(Y = k)} \mathbb{P}_s(Y = k \mid \boldsymbol{x}) \log \widehat{\mathbb{P}}_t(Y = k \mid \boldsymbol{z}). \quad (7)$$

According to Jensen's inequality, Eq. (7) attains its minimum value if and only if $\widehat{\mathbb{P}}_t(Y = k \mid \boldsymbol{x}) \mathbb{P}_s(Y = k) \propto \mathbb{P}_s(Y = k \mid \boldsymbol{x})$ for $k \in [C]$. Hence, Eq. (7) can be replaced as follows:

$$J = -\sum_{k \in [C]} \frac{\mathbb{P}(Y = k)}{\mathbb{P}_s(Y = k)} \mathbb{P}_s(Y = k \mid \boldsymbol{x}) \log \widehat{\mathbb{P}}(Y = k \mid \boldsymbol{z}), \quad (8)$$

where $\widehat{\mathbb{P}}(Y = y \mid \boldsymbol{z}) = \frac{\widehat{\mathbb{P}}_t(Y = y \mid \boldsymbol{z}) \mathbb{P}(Y = y)}{\sum_{k \in [C]} \widehat{\mathbb{P}}_t(Y = k \mid \boldsymbol{z}) \mathbb{P}(Y = k)}$ and $\mathbb{P}(Y)$ is an arbitrarily label distribution. Eq. (8) can be seen as an extension of sample reweighting [52] and logit adjustment [44] using probabilistic labels rather than discrete labels when $\mathbb{P}(Y)$ is specified as $\mathbb{P}_t(Y)$ and $\mathbb{P}_s(Y)$, respectively. Besides, Eq. (8) presents a unified framework that consolidates existing long-tail learning methods by estimating $\widehat{\mathbb{P}}(\boldsymbol{z} \mid Y = y)$ or $\widehat{\mathbb{P}}_t(Y = y \mid \boldsymbol{z})$ in different ways. In the following, we discuss three ways to estimate these terms.

**Method 1:** Explicitly specify $\widehat{\mathbb{P}}(\boldsymbol{z} \mid Y = y)$ as a prior distribution such as vMF distribution [34] and Wrapped Cauchy Distribution [27].

**Method 2:** Approximate $\widehat{\mathbb{P}}(\boldsymbol{z} \mid Y = y)$ using a learnable linear classifier. Let $\{\boldsymbol{w}_i, b_i\}_{i=1}^C$ denote the parameters of a linear layer, which is followed by a softmax to obtain the normalized probability:

$$\widehat{\mathbb{P}}(\boldsymbol{z} \mid Y = y) \propto \widehat{\mathbb{P}}_t(Y = y \mid \boldsymbol{z}) = \frac{\exp\left(\boldsymbol{z}^\top \boldsymbol{w}_y + b_y\right)}{\sum_{k \in [C]} \exp\left(\boldsymbol{z}^\top \boldsymbol{w}_k + b_k\right)}. \quad (9)$$

Table 1: A unified view of popular long-tail learning methods from our framework. "–" means that this method does not involve this issue and "×" indicates that the method has not resolved the issue.

| Method | Density estimation | Label distribution shift | Mini-batch computation of Eq. (10) |
|---|---|---|---|
| BALMS [51] | Linear layer | Reweighting | – |
| LA [44] | Linear layer | Logit adjustment | – |
| BCL [73] | Gaussian kernel | Reweighting | Class-wise center |
| GML [57] | Gaussian kernel | Logit adjustment | Class-wise queue |
| KCL [30] | Gaussian kernel | Balanced resampling | × |
| PaCo [15] | Gaussian kernel | Logit adjustment | Class-wise center |
| Proco [20] | Gaussian kernel | Logit adjustment | Class-wise vMF distribution |
| T-vMF [34] | T-vMF distribution | Logit adjustment | – |
| WCDAS [27] | Wrapped Cauchy distribution | Logit adjustment | – |

**Method 3:** Approximate $\widehat{\mathbb{P}}(z \mid Y = y)$ via the Gaussian kernel. A new sample from class $y$ should be closer to all samples within class $y$ and away from samples from other classes. Using the expected similarity among all samples within the class to measure distance, we derive:

$$\widehat{\mathbb{P}}(z \mid Y = y) \propto \widehat{\mathbb{P}}_t(Y = y \mid z) = \frac{\mathbb{E}_{x' \sim \mathbb{P}(\cdot|Y=y)} \left[ \kappa\left(z_x, z_{x'}\right) \right]}{\sum_{k \in [C]} \mathbb{E}_{x' \sim \mathbb{P}(\cdot|Y=k)} \left[ \kappa\left(z_x, z_{x'}\right) \right]}, \tag{10}$$

where $\kappa(\cdot, \cdot)$ represents the similarity between two samples, when we use Gaussian kernel $\kappa(z_x, z_{x'}) = \exp(z_x \cdot z_{x'})$ and approximate expectation through empirical batch $\mathcal{B} = \cup_{k \in [C]} \mathcal{B}_k$, that is $\mathbb{E}_{x' \sim \mathbb{P}(\cdot|Y=y)}[\kappa(z_x, z_{x'})] \approx \frac{1}{|\mathcal{B}_y|} \sum_{x' \in \mathcal{B}_y} \exp(z_x \cdot z_{x'})$, Eq. (10) can be instantiated as:

$$\widehat{\mathbb{P}}_t(Y = y \mid z) = \frac{\frac{1}{|\mathcal{B}_y|-1} \sum_{x' \in \mathcal{B}_y \setminus \{x\}} \exp\left(z_x \cdot z_{x'}\right)}{\sum_{k \in [C]} \frac{1}{|\mathcal{B}_k|} \sum_{x' \in \mathcal{B}_k} \exp\left(z_x \cdot z_{x'}\right)}. \tag{11}$$

Interestingly, we observe that Eq. (11) resembles class-balanced contrastive loss. In the appendix, we also show that the Gaussian kernel approximation is identical to conventional supervised contrastive learning if the training data are class-balanced.

Notably, to ensure the computability of $\mathbb{E}_{x' \sim \mathbb{P}(\cdot|Y=y)}[\kappa(z_x, z_{x'})]$ in Eq. (10), it is essential to ensure that samples are available from each class. Existing methods address this by class-wise queues, class-wise centers, or class-wise vMF distribution, details of which are provided in the appendix.

Based on the above three density approximation methods, we find that many recent proposals in long-tail learning can be derived from our framework. In Table 1, we summarize existing methods based on the way they estimate the density, tackle the training/test label shift, and guarantee the computation of Eq. (10) in mini-batch training.

## 3 CCL: Continuous Contrastive Learning

In this section, we introduce the proposed algorithm CCL, which extends the class-balanced contrastive learning presented in Eq. (8) with Gaussian kernel estimation in Eq. (11) to LTSSL.

### 3.1 Problem setup of long-tailed semi-supervised learning

Let $P_l$ and $P_u$ denote the joint distribution $(\mathcal{X}, \mathcal{Y})$ for labeled data and unlabeled data, respectively. We denote by $\mathbb{P}_l$ and $\mathbb{P}_u$ the corresponding probability density (or mass) functions. We possess a labeled dataset $\{(x_i^l, y_i^l)\}_{i=1}^N$ of size $N$ and an unlabeled dataset $\{x_j^u\}_{j=1}^M$ of size $M$, where $x_i^l, x_j^u \in \mathbb{R}^d$. The proportion of labeled data from the entire dataset is $\eta = \frac{N}{M+N}$. Denote the number of labeled data for class $i$ as $N_i$, we have $N_1 \geq N_2 \geq \ldots \geq N_C$ if the classes are sorted by cardinality in decreasing order. The imbalance ratio of labeled data is given by $\gamma_l = \frac{N_1}{N_C}$, while the distribution of the label of the unlabeled data and its imbalance ratio $\gamma_u$ are unknown. The components of CCL include a feature extractor, two linear classifiers $f_s(\cdot), f_b(\cdot)$ and a contrastive feature projection head $g(\cdot)$.

## 3.2 Balanced classifier training with estimated class prior

We develop our method based on FixMatch [56] following previous works [48, 64], and its objective is: $\widehat{\mathcal{L}}_{\text{ssl}} = \widehat{\mathcal{L}}_l + \widehat{\mathcal{L}}_u$, where $\widehat{\mathcal{L}}_l$ is a traditional cross-entropy loss. For unlabeled data, the method operates by first generating pseudo-labels for unlabeled data using the model's predictions and selecting unlabeled data whose predicted maximum confidence is higher than a predefined threshold. The consistency regularizer $\widehat{\mathcal{L}}_u$ is then applied to two views of each selected sample.

**Balanced FixMatch for LTSSL.** First, since the labeled data follow a long-tailed distribution, which we denote as $\boldsymbol{\pi}^l$, $\widehat{\mathcal{L}}_l$ needs to be adjusted by logit adjustment [44] via Eq. (8):

$$\widehat{\mathcal{L}}_l(\boldsymbol{x}^l, y^l) = -\log \frac{\widehat{\mathbb{P}}_t\left(Y = y^l \mid \boldsymbol{x}^l; f_b\right) \pi_{y^l}^l}{\sum_{k \in [C]} \widehat{\mathbb{P}}_t\left(Y = k \mid \boldsymbol{x}^l; f_b\right) \pi_k^l}. \tag{12}$$

Second, FixMatch is prone to fit wrong pseudo-labels with high predictive confidence during training [62, 3]. However, since the unlabeled data label distribution is inaccessible, pseudo-labels generated by the classifier may be sub-optimal for its adherence to a uniform distribution. By Bayes' theorem, with given estimated unlabeled data label distribution $\widehat{\boldsymbol{\pi}}^u$, the "post-adjusted" model outputs for sample $\boldsymbol{x}^u$ can be formulated as:

$$\widehat{\mathbb{P}}_u\left(Y = y \mid \boldsymbol{x}^u; f_b\right) = \frac{\widehat{\mathbb{P}}_t\left(Y = y \mid \boldsymbol{x}^u; f_b\right) \widehat{\pi}_y^u}{\sum_{k \in [C]} \widehat{\mathbb{P}}_t\left(Y = k \mid \boldsymbol{x}^u; f_b\right) \widehat{\pi}_k^u}, \tag{13}$$

Given pseudo-label $\widehat{y} = \arg \max_{k \in [C]} \widehat{\mathbb{P}}_u\left(Y = k \mid \mathcal{A}_w(\boldsymbol{x}^u); f_b\right)$, where $\mathcal{A}_w(\cdot)$ denotes the weak data augmentation, $\widehat{\mathcal{L}}_u$ is rewritten as:

$$\widehat{\mathcal{L}}_u(\boldsymbol{x}^u, \widehat{y}) = -\mathcal{M}(\boldsymbol{x}^u) \log \frac{\widehat{\mathbb{P}}_t\left(Y = \widehat{y} \mid \mathcal{A}_s(\boldsymbol{x}^u); f_b\right) \widehat{\pi}_{\widehat{y}}^u}{\sum_{k \in [C]} \widehat{\mathbb{P}}_t\left(Y = k \mid \mathcal{A}_s(\boldsymbol{x}^u); f_b\right) \widehat{\pi}_k^u}, \tag{14}$$

where $\mathcal{M}(\cdot)$ denotes the sample mask to select reliable pseudo-labels. We progressively update $\widehat{\boldsymbol{\pi}}^u$ using the exponential moving average (EMA) for each mini-batch by $\widehat{\pi}_y^u = (1 - \alpha)\widehat{\pi}_y^u + \frac{\alpha}{|\mathcal{B}|} \sum_{\boldsymbol{x}^u \in \mathcal{B}} \widehat{\mathbb{P}}_u\left(Y = y \mid \boldsymbol{x}^u; f_b\right)$, where $\alpha$ is a momentum updating parameter and $\mathcal{B}$ denotes an unlabeled data subset. Directly using confidence selection can lead to a selected $\mathcal{B}$ with poor calibration due to model overconfidence [41, 45]. Thus, the energy score [36] is adopted to filter out reliable unlabeled data, which is defined as $E(\boldsymbol{x}) = -T \cdot \log \sum_{k \in [C]} e^{f_k(\boldsymbol{x})/T}$, where $T$ is the temperature and $f(\boldsymbol{x})$ denotes the logits of $\boldsymbol{x}$. We select reliable unlabeled data by $\mathcal{M}^E(\boldsymbol{x}^u) := \mathbb{I}(E(\boldsymbol{x}^u) \leq \zeta)$ using a predefined threshold $\zeta$, and construct $\mathcal{B} = \{\boldsymbol{x} \mid \boldsymbol{x} \in \mathcal{B}^u \wedge \mathcal{M}^E(\boldsymbol{x}) \neq 0\}$ for the estimation of $\widehat{\boldsymbol{\pi}}_u$.

**Dual-branches training.** Based on the observation that class-balanced training can be harmful to representation learning, previous works [38, 64] have utilized another branch of the network for standard training. In contrast to the balanced branch, the standard branch, denoted as $f_s(\cdot)$, optimizes the cross-entropy loss without employing logit adjustment to fit the original training data distribution. We fuse the predictions of balanced and standard branches to reduce the confirmation bias of pseudo-labels by:

$$\widehat{\mathbb{P}}^{\text{cls}}\left(Y = y \mid \boldsymbol{x}^u\right) = \frac{1}{2}\widehat{\mathbb{P}}_u\left(Y = y \mid \boldsymbol{x}^u; f_b\right) + \frac{1}{2} \frac{\widehat{\mathbb{P}}\left(Y = y \mid \boldsymbol{x}^u; f_s\right) \widehat{\pi}_y^*}{\sum_{k \in [C]} \widehat{\mathbb{P}}\left(Y = k \mid \boldsymbol{x}^u; f_s\right) \widehat{\pi}_k^*}. \tag{15}$$

The rationale behind the equation is that the standard branch necessitates the elimination of imbalanced label prior and then compensates for unlabeled label prior when predicting pseudo-labels, which is achieved by defining $\widehat{\boldsymbol{\pi}}^* = \frac{\widehat{\boldsymbol{\pi}}^u}{\boldsymbol{\pi}^l + \widehat{\boldsymbol{\pi}}^u}$. Overall, the classification loss $\widehat{\mathcal{L}}_{\text{cls}}$ is the combination of losses for learning $f_s(\cdot)$ and $f_b(\cdot)$.

## 3.3 Continuous contrastive loss with reliable pseudo-labels

Apart from the classification loss and consistency regularizer, we aim to improve the quality of representations by extending the framework presented in Section 2 to LTSSL. To achieve the adaptation

of our framework, a primary obstacle must be addressed. The challenge arises from the unknown ground-truth label $\mathbb{P}_u(Y = y \mid \boldsymbol{x})$ for unlabeled data, which results in the calculation of Eq. (8) infeasible. We propose to utilize the continuous pseudolabel $\widehat{\mathbb{P}}^{\mathrm{cls}}(Y = y \mid \boldsymbol{x}^u)$ as derived from the classifier in Eq. (15). Furthermore, $\mathbb{E}_{\boldsymbol{x}' \sim \mathbb{P}(\cdot \mid Y=y)}[\kappa(\boldsymbol{z}_{\boldsymbol{x}}, \boldsymbol{z}_{\boldsymbol{x}'})]$ in Eq. (10) can be extended to unlabeled data and approximated by an empirical data subset $\mathcal{B}$:

$$\mathbb{E}_{\boldsymbol{x}' \sim \mathbb{P}(\cdot \mid Y=y)}[\kappa(\boldsymbol{z}_{\boldsymbol{x}}, \boldsymbol{z}_{\boldsymbol{x}'})] \approx \frac{\sum_{\boldsymbol{x}' \in \mathcal{B}} \kappa(\boldsymbol{z}_{\boldsymbol{x}}, \boldsymbol{z}_{\boldsymbol{x}'}) \widehat{\mathbb{P}}^{\mathrm{cls}}(Y = y \mid \boldsymbol{x}')}{\sum_{\boldsymbol{x}' \in \mathcal{B}} \widehat{\mathbb{P}}^{\mathrm{cls}}(Y = y \mid \boldsymbol{x}')}. \tag{16}$$

Plugging Eq. (16) into Eq. (10), we obtain the continuous pseudo-label $\widehat{\mathbb{P}}_t(Y = y \mid \boldsymbol{x}^u; \mathcal{B})$ for $\boldsymbol{x}^u$. Similar to Eq. (14), logit adjustment is used to handle label shift of unlabeled data by Bayes' theorem, and we can obtain:

$$\widehat{\mathcal{L}}_{\mathrm{rpl}} = -\sum_{k \in [C]} \widehat{\mathbb{P}}^{\mathrm{cls}}(Y = k \mid \boldsymbol{x}^u) \cdot \log \widehat{\mathbb{P}}_u(Y = k \mid \boldsymbol{x}^u; \mathcal{B}), \tag{17}$$

where $\widehat{\mathbb{P}}_u(Y = y \mid \boldsymbol{x}^u; \mathcal{B}) = \frac{\widehat{\mathbb{P}}_t(Y=y \mid \boldsymbol{x}^u; \mathcal{B}) \cdot \widehat{\pi}_y^u}{\sum_{k \in [C]} \widehat{\mathbb{P}}_t(Y=k \mid \boldsymbol{x}^u; \mathcal{B}) \cdot \widehat{\pi}_k^u}$. So far, the generalized framework using continuous pseudo-labels for LTSSL is derived in Eq. (17), the critical issue is how to filter out a reliable unlabeled data subset $\mathcal{B}^u$ such that the posterior estimation $\widehat{\mathbb{P}}^{\mathrm{cls}}(Y = y \mid \boldsymbol{x})$ in Eq. (16) is calibrated. Similarly, directly using confidence selection may lead to model overconfidence. To mitigate the confirmation bias in pseudo-labels produced by the learned classifier, we propose using the energy score for data selection to ensure model calibration [26]. Combining with labeled data, the loss $\widehat{\mathcal{L}}_{\mathrm{rpl}}$ is obtained with $\mathcal{B} = \{\boldsymbol{x} \mid \boldsymbol{x} \in \mathcal{B}^l \vee (\boldsymbol{x} \in \mathcal{B}^u \wedge \mathcal{M}^E(\boldsymbol{x}) \neq 0)\}$.

### 3.4 Continuous contrastive loss with smoothed pseudo-labels

To further mitigate the impact of inaccurate pseudo-labels $\widehat{\mathbb{P}}^{\mathrm{cls}}(Y = y \mid \boldsymbol{x}^u)$, we derive a complementary contrastive loss with smoothed pseudo-labels. Specifically, we propose aligning the representations of two views of a sample by imposing the weak-strong consistency regularization:

$$\widehat{\mathcal{L}}_{\mathrm{spl}} = -\sum_{k \in [C]} \widehat{\mathbb{P}}(Y = k \mid \mathcal{A}_w(\boldsymbol{x}^u)) \log \widehat{\mathbb{P}}(Y = k \mid \mathcal{A}_s(\boldsymbol{x}^u)). \tag{18}$$

In this part, we aim to derive $\widehat{\mathbb{P}}(Y = y \mid \boldsymbol{x}^u)$ by propagating labels from nearby samples in the contrastive space. On the one hand, we take labeled data for $\mathcal{B}$ in Eq. (11) and construct the posterior for unlabeled data. Logit adjustment is employed for tackling label shift of unlabeled data:

$$\widehat{\mathbb{P}}(Y = y \mid \boldsymbol{x}^u; \mathcal{B}^l) = \frac{\frac{1}{|\mathcal{B}_y|} \sum_{\boldsymbol{x}' \in \mathcal{B}_y} \kappa(\boldsymbol{z}_{\boldsymbol{x}^u}, \boldsymbol{z}_{\boldsymbol{x}'}) \cdot \widehat{\pi}_y^u}{\sum_{k \in [C]} \frac{1}{|\mathcal{B}_k|} \sum_{\boldsymbol{x}' \in \mathcal{B}_k} \kappa(\boldsymbol{z}_{\boldsymbol{x}^u}, \boldsymbol{z}_{\boldsymbol{x}'}) \cdot \widehat{\pi}_k^u}. \tag{19}$$

Eq. (19) can be viewed as a process of propagating labels from labeled data to unlabeled data. On the other hand, we consider label propagation within unlabeled data, i.e., an unlabeled batch $\mathcal{B}^u$ is used to estimate $\widehat{\mathbb{P}}(Y = y \mid \boldsymbol{x}^u; \mathcal{B})$. Assuming there is a sufficient amount of unlabeled data, we have $\frac{1}{|\mathcal{B}^u|} \sum_{\boldsymbol{x}^u \in \mathcal{B}^u} \widehat{\mathbb{P}}(Y = y \mid \boldsymbol{x}^u; \mathcal{B}^u) \approx \widehat{\pi}_y^u$, hence the posterior can be approximated as:

$$\widehat{\mathbb{P}}(Y = y \mid \boldsymbol{x}^u; \mathcal{B}^u) \approx \frac{\sum_{\boldsymbol{x}' \in \mathcal{B}^u} \kappa(\boldsymbol{z}_{\boldsymbol{x}^u}, \boldsymbol{z}_{\boldsymbol{x}'}) \widehat{\mathbb{P}}(Y = y \mid \boldsymbol{x}'; \mathcal{B}^u)}{\sum_{\boldsymbol{x}' \in \mathcal{B}^u} \kappa(\boldsymbol{z}_{\boldsymbol{x}^u}, \boldsymbol{z}_{\boldsymbol{x}'})}. \tag{20}$$

Let $\mathbb{P}(Y \mid \boldsymbol{X}; \mathcal{B})$ represent a matrix stacked by $[\mathbb{P}(Y = 1 \mid \boldsymbol{x}), \ldots, \mathbb{P}(Y = C \mid \boldsymbol{x})]^\top$ of $\boldsymbol{x}$ from $\mathcal{B}$, we can rewrite Eq. (20) in the form of matrix multiplication: $\widehat{\mathbb{P}}(Y \mid \boldsymbol{X}^u; \mathcal{B}^u) = \boldsymbol{G} \cdot \widehat{\mathbb{P}}(Y \mid \boldsymbol{X}^u; \mathcal{B}^u)$, where $\boldsymbol{G}$ is a similarity matrix and $G_{ij} = \frac{\kappa(\boldsymbol{z}_{\boldsymbol{x}_i}, \boldsymbol{z}_{\boldsymbol{x}_j})}{\sum_{\boldsymbol{x}_j \in \mathcal{B}_u} \kappa(\boldsymbol{z}_{\boldsymbol{x}_i}, \boldsymbol{z}_{\boldsymbol{x}_j})}$. It can be interpreted that similar samples possess similar labels. By aggregating the predictions of labeled data and unlabeled data with a fixed hyperparameter $\beta$, we obtain:

$$\begin{aligned} \widehat{\mathbb{P}}(Y \mid \boldsymbol{X}^u) &= \beta \boldsymbol{G} \cdot \widehat{\mathbb{P}}(Y \mid \boldsymbol{X}^u) + (1 - \beta) \widehat{\mathbb{P}}(Y \mid \boldsymbol{X}^u; \mathcal{B}^l) \\ &\Rightarrow \widehat{\mathbb{P}}(Y \mid \boldsymbol{X}^u) = (1 - \beta)(\boldsymbol{I} - \beta \boldsymbol{G})^{-1} \cdot \widehat{\mathbb{P}}(Y \mid \boldsymbol{X}^u; \mathcal{B}^l). \end{aligned} \tag{21}$$

Subsequently, Eq. (21) can be plugged into Eq. (18) for calculating $\widehat{\mathcal{L}}_{\mathrm{spl}}$. To sum up, the total objective of CCL is:

$$\widehat{\mathcal{L}}_{\mathrm{total}} = \lambda_1 \widehat{\mathcal{L}}_{\mathrm{cls}} + (1 - \lambda_1) \widehat{\mathcal{L}}_{\mathrm{rpl}} + \lambda_2 \widehat{\mathcal{L}}_{\mathrm{spl}} \tag{22}$$

where $\lambda_1$ and $\lambda_2$ are two hyperparameters.

Table 2: Test accuracy in `consistent` setting on CIFAR10-LT and CIFAR100-LT datasets. The best results are in **bold**.

| | CIFAR10-LT | | | | CIFAR100-LT | | | |
| | $\gamma = \gamma_l = \gamma_u = 100$ | | $\gamma = \gamma_l = \gamma_u = 150$ | | $\gamma = \gamma_l = \gamma_u = 10$ | | $\gamma = \gamma_l = \gamma_u = 20$ | |
| Algorithm | $N_1 = 500$ $M_1 = 4000$ | $N_1 = 1500$ $M_1 = 3000$ | $N_1 = 500$ $M_1 = 4000$ | $N_1 = 1500$ $M_1 = 3000$ | $N_1 = 50$ $M_1 = 400$ | $N_1 = 150$ $M_1 = 300$ | $N_1 = 50$ $M_1 = 400$ | $N_1 = 150$ $M_1 = 300$ |
|---|---|---|---|---|---|---|---|---|
| Supervised | 47.3 ±0.95 | 61.9 ±0.41 | 44.2 ±0.33 | 58.2 ±0.29 | 29.6 ±0.57 | 46.9 ±0.22 | 25.1 ±1.14 | 41.2 ±0.15 |
| w/ LA [44] | 53.3 ±0.44 | 70.6 ±0.21 | 49.5 ±0.40 | 67.1 ±0.78 | 30.2 ±0.44 | 48.7 ±0.89 | 26.5 ±1.31 | 44.1 ±0.42 |
| FixMatch [56] | 67.8 ±1.13 | 77.5 ±1.32 | 62.9 ±0.36 | 72.4 ±1.03 | 45.2 ±0.55 | 56.5 ±0.06 | 40.0 ±0.96 | 50.7 ±0.25 |
| w/ DARP [33] | 74.5 ±0.78 | 77.8 ±0.63 | 67.2 ±0.32 | 73.6 ±0.73 | 49.4 ±0.20 | 58.1 ±0.44 | 43.4 ±0.87 | 52.2 ±0.66 |
| w/ CReST+ [63] | 76.3 ±0.86 | 78.1 ±0.42 | 67.5 ±0.45 | 73.7 ±0.34 | 44.5 ±0.94 | 57.4 ±0.18 | 40.1 ±1.28 | 52.1 ±0.21 |
| w/ DASO [48] | 76.0 ±0.37 | 79.1 ±0.75 | 70.1 ±1.81 | 75.1 ±0.77 | 49.8 ±0.24 | 59.2 ±0.35 | 43.6 ±0.09 | 52.9 ±0.42 |
| FixMatch + LA [44] | 75.3 ±2.45 | 82.0 ±0.36 | 67.0 ±2.49 | 78.0 ±0.91 | 47.3 ±0.42 | 58.6 ±0.36 | 41.4 ±0.93 | 53.4 ±0.32 |
| w/ DARP [33] | 76.6 ±0.92 | 80.8 ±0.62 | 68.2 ±0.94 | 76.7 ±1.13 | 50.5 ±0.78 | 59.9 ±0.32 | 44.4 ±0.65 | 53.8 ±0.43 |
| w/ CReST+ [63] | 76.7 ±1.13 | 81.1 ±0.57 | 70.9 ±1.18 | 77.9 ±0.71 | 44.0 ±0.21 | 57.1 ±0.55 | 40.6 ±0.55 | 52.3 ±0.20 |
| w/ DASO [48] | 77.9 ±0.88 | 82.5 ±0.08 | 70.1 ±1.68 | 79.0 ±2.23 | 50.7 ±0.51 | 60.6 ±0.71 | 44.1 ±0.61 | 55.1 ±0.72 |
| FixMatch + ABC [38] | 78.9 ±0.82 | 83.8 ±0.36 | 66.5 ±0.78 | 80.1 ±0.45 | 47.5 ±0.18 | 59.1 ±.21 | 41.6 ±0.83 | 53.7 ±0.55 |
| w/ DASO [48] | 80.1 ±1.16 | 83.4 ±0.31 | 70.6 ±0.80 | 80.4 ±0.56 | 50.2 ±0.62 | 60.0 ±0.32 | 44.5 ±0.25 | 55.3 ±0.53 |
| FixMatch + ACR [64] | 81.6 ±0.19 | 84.1 ±0.39 | 77.0 ±1.19 | 80.9 ±0.22 | 51.1 ±0.32 | 61.0 ±0.41 | 44.3 ±0.21 | 55.2 ±0.28 |
| FixMatch + CPE [43] | 80.7 ±0.96 | 84.4 ±0.29 | 76.8 ±0.53 | 82.3 ±0.34 | 50.3 ±0.34 | 59.8 ±0.16 | 43.8 ±0.28 | 55.6 ±0.15 |
| FixMatch + **CCL** | **84.5** ±0.38 | **86.2** ±0.35 | **81.5** ±0.99 | **84.0** ±0.21 | **53.5** ±0.49 | **63.5** ±0.39 | **46.8** ±0.45 | **57.5** ±0.16 |

Table 3: Test accuracy under inconsistent setting ($\gamma_l \neq \gamma_u$) on CIFAR10-LT and STL10-LT datasets. $\gamma_l = 100$ for CIFAR10-LT, and 10 and 20 for STL10-LT dataset. The best results are in **bold**.

| | CIFAR10-LT ($\gamma_l \neq \gamma_u$) | | | | STL10-LT ($\gamma_u$ =N/A) | | | |
| | $\gamma_u = 1$ (uniform) | | $\gamma_u = 1/100$ (reversed) | | $\gamma_l = 10$ | | $\gamma_l = 20$ | |
| Algorithm | $N_1 = 500$ $M_1 = 4000$ | $N_1 = 1500$ $M_1 = 3000$ | $N_1 = 500$ $M_1 = 4000$ | $N_1 = 1500$ $M_1 = 3000$ | $N_1 = 150$ $M = 100k$ | $N_1 = 450$ $M = 100k$ | $N_1 = 150$ $M = 100k$ | $N_1 = 450$ $M = 100k$ |
|---|---|---|---|---|---|---|---|---|
| FixMatch | 73.0 ±3.81 | 81.5 ±1.15 | 62.5 ±0.94 | 71.8 ±1.70 | 56.1 ±2.32 | 72.4 ±0.71 | 47.6 ±4.87 | 64.0 ±2.27 |
| w/ DARP [33] | 82.5 ±0.75 | 84.6 ±0.34 | 70.1 ±0.22 | 80.0 ±0.93 | 66.9 ±1.66 | 75.6 ±0.45 | 59.9 ±2.17 | 72.3 ±0.60 |
| w/ CReST [63] | 83.2 ±1.67 | 87.1 ±0.28 | 70.7 ±2.02 | 80.8 ±0.39 | 61.7 ±2.51 | 71.6 ±1.17 | 57.1 ±3.67 | 68.6 ±0.88 |
| w/ CReST+ [63] | 82.2 ±1.53 | 86.4 ±0.42 | 62.9 ±1.39 | 72.9 ±2.00 | 61.2 ±1.27 | 71.5 ±0.96 | 56.0 ±3.19 | 68.5 ±1.88 |
| w/ DASO [48] | 86.6 ±0.84 | 88.8 ±0.59 | 71.0 ±0.95 | 80.3 ±0.65 | 70.0 ±1.19 | 78.4 ±0.80 | 65.7 ±1.78 | 75.3 ±0.44 |
| w/ ACR [64] | 92.1 ±0.18 | 93.5 ±0.11 | 85.0 ±0.99 | 89.5 ±0.17 | 77.1 ±0.24 | 83.0 ±0.32 | 75.1 ±0.70 | 81.5 ±0.25 |
| w/ CPE [43] | 92.3 ±0.17 | 93.3 ±0.21 | 84.8 ±0.88 | 89.3 ±0.11 | 73.1 ±0.47 | 83.3 ±0.14 | 69.6 ±0.20 | 81.7 ±0.34 |
| w/ **CCL** | **93.1** ±0.21 | **93.9** ±0.12 | **85.0** ±0.70 | **89.8** ±0.31 | **79.1** ±0.43 | **84.8** ±0.15 | **77.1** ±0.33 | **83.1** ±0.18 |

## 4 Experiments

In this section, we conducted comprehensive experiments to verify the effectiveness of the proposed continuous contrastive learning method (CCL) on CIFAR10-LT, CIFAR100-LT, STL10-LT, and ImageNet-127 [29, 18] datasets. To simulate real-world unlabeled data, we tested our method on diverse label distributions of unlabeled data. Due to limited space, we defer the detailed experimental settings to the appendix.

### 4.1 Results on CIFAR10/100-LT and STL10-LT

For `consistent` ($\gamma_l = \gamma_u$) setting, results are presented in Table 2. From the results, we can see that CCL consistently outperforms all comparison methods by a large margin. In particular, CCL improves the previous state-of-the-art approach ACR by 2.8% on average. This verifies that the representations learned by our proposed contrastive losses are more discriminative because both CCL and ACR utilize a dual-branch network.

For `inconsistent` ($\gamma_l \neq \gamma_u$) setting, we present the results in Table 3 and Table 4. Following prior works, we compare all methods using unlabeled data following `uniform` and `reversed` label distributions on CIFAR10/100-LT datasets. On the STL10-LT dataset, the underlying unlabeled data distribution is naturally inaccessible. In general, CCL achieves the best results in all settings. Particularly, CCL obtains an average performance gain of 1.6% over ACR without using predefined anchor distributions. The results indicate that our method is able to accurately estimate the unlabeled data distribution with calibrated pseudo-labels.

## 4.2 Results on ImageNet-127

ImageNet-127 is a naturally long-tailed dataset and has been used to test LTSSL methods in the recent literature [22, 64]. Following previous works, we downsample the original images to smaller sizes of 32×32 or 64×64 pixels using the box method from the Pillow library and randomly select 10% training samples to construct the labeled data. Learning discriminative representations and a balanced classifier is essential to achieve high performance. From the results in Table 5, we can see that CCLachieves superior results for image sizes of 32×32 and 64×64. It outperforms ACR by 4.3% and 4.2% in test accuracy.

Table 4: Test accuracy on CIFAR100-LT in `uniform` and `reversed` settings. The best results are in **bold**.

| Algorithm | $\gamma_u = 1$ (uniform) | | $\gamma_u = 1/10$ (reversed) | |
| --- | --- | --- | --- | --- |
| | $N_1 = 50$ $M_1 = 400$ | $N_1 = 150$ $M_1 = 300$ | $N_1 = 50$ $M_1 = 400$ | $N_1 = 150$ $M_1 = 300$ |
| FixMatch | 45.5 ±0.71 | 58.1 ±0.72 | 44.2 ±0.43 | 57.3 ±0.19 |
| w/ DARP [33] | 43.5 ±0.95 | 55.9 ±0.32 | 36.9 ±0.48 | 51.8 ±0.92 |
| w/ CReST [63] | 43.5 ±0.30 | 59.2 ±0.25 | 39.0 ±1.11 | 56.4 ±0.62 |
| w/ CReST+ [63] | 43.6 ±1.60 | 58.7 ±0.16 | 39.1 ±0.77 | 56.4 ±0.78 |
| w/ DASO [48] | 53.9 ±0.66 | 61.8 ±0.98 | 51.0 ±0.19 | 60.0 ±0.31 |
| w/ ACR [64] | 57.9 ±0.56 | 65.8 ±0.91 | 51.7 ±0.22 | 63.3 ±0.17 |
| w/ CCL | **59.8** ±0.28 | **67.9** ±0.70 | **54.4** ±0.14 | **64.7** ±0.22 |

Table 5: Test accuracy on ImageNet-127. The best results are in **bold**.

| Algorithm | $32 \times 32$ | $64 \times 64$ |
| --- | --- | --- |
| FixMatch | 29.7 | 42.3 |
| w/ DARP [33] | 30.5 | 42.5 |
| w/ DARP+cRT [33] | 39.7 | 51.0 |
| w/ CReST+ [63] | 32.5 | 44.7 |
| w/ CReST++LA [63] | 40.9 | 55.9 |
| w/ CoSSL [22] | 43.7 | 53.9 |
| w/ TRAS [66] | 46.2 | 54.1 |
| w/ ACR [64] | 57.2 | 63.6 |
| w/ CCL | **61.5** | **67.8** |

## 4.3 Comprehensive evaluation of the proposed method

**Understanding of balanced Fixmatch and dual-branch.** First, balanced Fixmatch can be viewed as a separate EM algorithm process [17, 21], where the E-step involves estimating suitable pseudo-labels for unlabeled data through $\widehat{\pi}^u$, and the M-step updates the model and obtains a new $\widehat{\pi}^u$. As can be seen in Table 6, balanced Fixmatch achieves performance comparable to the recent state-of-the-art method ACR. Furthermore, dual-branch significantly enhances the performance of data under highly skewed long-tail distributions (`consistent` setting), with an averaged 1.5% improvement.

Table 6: Ablation studies of our proposed algorithm. "Con", "Uni", and "Rev" represent `consistent`, `uniform`, and `reversed`, respectively.

| Dual-branch | Reliable PL | Smoothed PL | Energy mask | CIFAR10-LT | | | CIFAR100-LT | | |
| --- | --- | --- | --- | --- | --- | --- | --- | --- | --- |
| | | | | Con | Uni | Rev | Con | Uni | Rev |
| | | | ✓ | 81.0 | 91.8 | 84.2 | 49.3 | 57.0 | 51.5 |
| ✓ | | | ✓ | 82.1 | 92.0 | 84.5 | 51.2 | 57.3 | 52.1 |
| ✓ | | ✓ | ✓ | 83.5 | 92.8 | 84.7 | 52.7 | 58.5 | 53.2 |
| ✓ | ✓ | | ✓ | 83.2 | 92.7 | 84.8 | 51.9 | 58.4 | 53.2 |
| ✓ | ✓ | ✓ | | 83.8 | 92.7 | 84.8 | 52.7 | 59.1 | 53.9 |
| ✓ | ✓ | ✓ | ✓ | 84.3 | 93.1 | 85.0 | 53.5 | 59.9 | 54.4 |

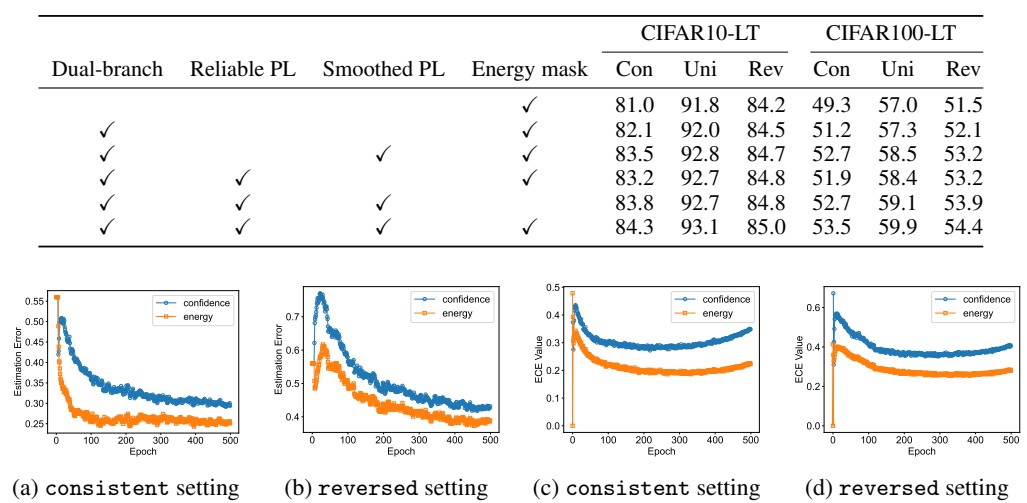

(a) `consistent` setting    (b) `reversed` setting    (c) `consistent` setting    (d) `reversed` setting

Figure 1: Comparison of class prior estimation error and ECE on CIFAR100-LT.

**How to estimate a relatively accurate $\widehat{\pi}_u$?** Our method for estimating $\widehat{\pi}_u$ is equivalent to MLLS [53], which is an EM process. Accurate estimation of $\widehat{\pi}_u$ is only achievable when the model is calibrated [25]. Since the confirmation bias is induced by self-training, using confidence selection may result in overconfident but wrong pseudo-labels and hurt the calibration [45, 41]. In contrast, the energy score leverages the probability density of the predictions, exhibiting reduced vulnerability to overconfidence [39]. Thus, we propose energy selection for a reliable unlabeled data subset on which the model is calibrated, thereby enabling the accurate estimation of $\widehat{\pi}_u$. We use expected

calibration error [26] (ECE) to assess model calibration. The tail of the curve of Figure 1c and 1d can be interpreted as overconfidence in false pseudo-labels caused by self-training. As can be seen in Figure 1a and 1b, the $L_1$ distance between the true class prior of unlabeled data and $\widehat{\pi}_u$, estimated from data subset selected using energy, is significantly smaller compared to when confidence is used for selection, inducing a more balanced classifier training.

**Continuous contrastive learning with reliable pseudo-labels.** We carried out a comparative experiment by removing the continuous reliable pseudo-labels loss. The results reflect an averaged $0.8\%$ drop on CIFAR10/100-LT, demonstrating its efficacy for learning high-quality representation. Moreover, we verified that the data subset filtered by energy selection obtains excellent model calibration. Figure 1c and 1d show energy achieves better calibration than confidence thresholding.

**Continuous contrastive learning with smoothed pseudo-labels.** Similarly, we conducted a comparative experiment by removing the continuous smoothed pseudo-labels loss. As can be seen in Table 6, the performance decreases in all three settings on CIFAR10/100-LT datasets, showing the necessity for a consistency regularization constraint for feature alignment within the contrastive learning space.

### 4.4 Results under more class distributions

Similar to ACR, to evaluate our method's effectiveness under more imbalanced settings, we conducted further experiments on CIFAR100-LT, maintaining a fixed $\gamma_l = 20$ and adjusting the imbalance ratio $\gamma_u$ of the unlabeled data from consistent to reversed. We set $N_1 = 50$ and $M_1 = 400$ (with $M_C = 400$ in the `reversed` scenario) and compared the results with ACR as shown in Figure 2. The results demonstrate that our method consistently outperforms ACR in all scenarios.

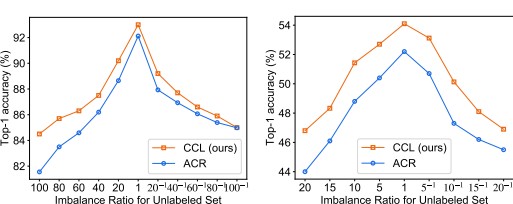

(a) CIFAR10-LT $\gamma_l = 100$ (b) CIFAR100-LT $\gamma_l = 20$

Figure 2: Generalize to more realistic LTSSL settings for ACR and CCL on CIFAR10/100-LT dataset in fixed $\gamma_l$ and various $\gamma_u$ settings.

## 5 Related Work

**Long-tailed learning (LTL).** Early strategies tackling LTL involve two aspects: resampling and reweighting. Resampling methods [9, 5, 8, 54] either undersample majority classes or oversample minority classes, which may result in information loss or overfitting. Reweighting methods [52, 16, 2, 12] assign different weights for each class or training sample. BBN [72] and Decoupling [31] claim that re-balancing can negatively impact representation. They propose a two-branch structure or a two-stage paradigm to address it. Logit adjustment methods [7, 44] learn larger margins for minority classes by obtaining optimal Bayesian classifiers. Recently, several methods [30, 15, 73, 20, 57] have been proposed to improve the representation learning based on supervised contrastive learning [32].

**Long-tailed semi-supervised learning (LTSSL).** Most semi-supervised learning (SSL) methods use unlabeled data by assigning pseudo-labels to unlabeled data [37, 6, 56, 71, 10] or aligning predictions of different views of the input by consistency regularization [59]. PAWS [4] leverages self-supervised representations derived from unlabeled data, and RoPAWS [46] further refines the model predictions using labeled data through kernel density estimation. However, most of these works assume a balanced class distribution of labeled and unlabeled data, which may be violated in real-world applications.

Recently, LTSSL has gained considerable attention due to its applicability in numerous real-life scenarios. Recent works mitigate pseudo-labels bias by distribution alignment or label refinement [33, 63, 69]. Some others focus on balanced classifier training to overcome long-tailed label distribution [38, 22, 66]. Regrettably, these methods simply assume an identical long-tailed distribution for labeled and unlabeled data, which may still be unrealistic. Considering the unknown unlabeled data distribution, which can be mismatched with the labeled distribution, DASO [48] mixes the outputs of linear and semantic classifiers to improve the quality of pseudo-labels. ACR [64] and CPE [43] refine consistency regularization or train multiple expert branches based on predefined anchor distributions. However, how to improve representation learning in LTSSL is ignored in most existing works.

# 6 Conclusion

This paper presents a probabilistic framework that unifies many recent methods in long-tail learning. Our framework is equivalent to supervised contrastive learning when approximating the class-conditional function using the Gaussian kernel. We further extend the contrastive learning objective to LTSSL based on continuous pseudo-labels to improve the learned representations. We utilize both reliable pseudo-labels generated by the model and smoothed pseudo-labels propagated from nearby samples to mitigate confirmation bias. Extensive experiments demonstrate that our proposed method achieves state-of-the-art performance in all settings. We hope that our work can motivate more research for LTSSL from the perspective of representation learning.

## Acknowledgments and Disclosure of Funding

This work was supported by the National Science Foundation of China (62206049, 62225602), and the Big Data Computing Center of Southeast University. We would like to thank anonymous reviewers for their constructive suggestions.

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

## A  Comparison with Supervised Contrastive Learning

As we have derived in Eq. (10) and use the Gaussian kernel density estimation in Eq. (11), if we simply assume the data are class-balanced, it simplifies to:

$$
\widehat{\mathbb{P}}(Y = y \mid \boldsymbol{z}) = \frac{\left(\frac{1}{|\mathcal{B}_y|-1} \sum_{\boldsymbol{x}' \in \mathcal{B}_y \setminus \{\boldsymbol{x}\}} \exp\left(\boldsymbol{z}_{\boldsymbol{x}} \cdot \boldsymbol{z}_{\boldsymbol{x}'}\right)\right) \mathbb{P}(Y = y)}{\sum_{k \in [C]} \left(\frac{1}{|\mathcal{B}_k|} \sum_{\boldsymbol{x}' \in \mathcal{B}_k} \exp\left(\boldsymbol{z}_{\boldsymbol{x}} \cdot \boldsymbol{z}_{\boldsymbol{x}'}\right)\right) \mathbb{P}(Y = k)}
$$

$$
= \frac{\sum_{\boldsymbol{x}' \in \mathcal{B}_y \setminus \{\boldsymbol{x}\}} \exp\left(\boldsymbol{z}_{\boldsymbol{x}} \cdot \boldsymbol{z}_{\boldsymbol{x}'}\right)}{\sum_{k \in [C]} \sum_{\boldsymbol{x}' \in \mathcal{B}_k} \exp\left(\boldsymbol{z}_{\boldsymbol{x}} \cdot \boldsymbol{z}_{\boldsymbol{x}'}\right)} = \frac{\sum_{\boldsymbol{x}' \in \mathcal{B}_y \setminus \{\boldsymbol{x}\}} \exp\left(\boldsymbol{z}_{\boldsymbol{x}} \cdot \boldsymbol{z}_{\boldsymbol{x}'}\right)}{\sum_{\boldsymbol{x}' \in \mathcal{B}} \exp\left(\boldsymbol{z}_{\boldsymbol{x}} \cdot \boldsymbol{z}_{\boldsymbol{x}'}\right)}.
\tag{23}
$$

The loss of supervised contrastive learning has two forms, i.e., $\widehat{\mathcal{L}}_{\mathrm{scl}}^{\mathrm{in}}$ and $\widehat{\mathcal{L}}_{\mathrm{scl}}^{\mathrm{out}}$, which is distinguished by the position of summation over positive samples at the $\log(\cdot)$. Thus, we get:

$$
\widehat{\mathcal{L}}_{\mathrm{scl}}^{\mathrm{in}}(\boldsymbol{x}, y) = -\log \frac{\sum_{\boldsymbol{x}' \in \mathcal{B}_y \setminus \{\boldsymbol{x}\}} \exp\left(\boldsymbol{z}_{\boldsymbol{x}} \cdot \boldsymbol{z}_{\boldsymbol{x}'}\right)}{\sum_{\boldsymbol{x}' \in \mathcal{B}} \exp\left(\boldsymbol{z}_{\boldsymbol{x}} \cdot \boldsymbol{z}_{\boldsymbol{x}'}\right)}
$$

$$
\propto -\log \frac{\frac{1}{|\mathcal{B}_y|-1} \sum_{\boldsymbol{x}' \in \mathcal{B}_y \setminus \{\boldsymbol{x}\}} \exp\left(\boldsymbol{z}_{\boldsymbol{x}} \cdot \boldsymbol{z}_{\boldsymbol{x}'}\right)}{\sum_{\boldsymbol{x}' \in \mathcal{B}} \exp\left(\boldsymbol{z}_{\boldsymbol{x}} \cdot \boldsymbol{z}_{\boldsymbol{x}'}\right)}
\tag{24}
$$

$$
\overset{\mathrm{Jensen}}{\leq} -\frac{1}{|\mathcal{B}_y| - 1} \sum_{\boldsymbol{x}' \in \mathcal{B}_y \setminus \{\boldsymbol{x}\}} \log \frac{\exp\left(\boldsymbol{z}_{\boldsymbol{x}} \cdot \boldsymbol{z}_{\boldsymbol{x}'}\right)}{\sum_{\boldsymbol{x}' \in \mathcal{B}} \exp\left(\boldsymbol{z}_{\boldsymbol{x}} \cdot \boldsymbol{z}_{\boldsymbol{x}'}\right)} = \widehat{\mathcal{L}}_{\mathrm{scl}}^{\mathrm{out}}(\boldsymbol{x}, y).
$$

which is consistent with the original paper's derivation.

## B  Analysis of Existing Long-Tail Learning Methods

As we have derived before, here we dive deep into the analysis of existing methods and demonstrate that they all belong to our unified framework.

### B.1  Discussion of Gaussian kernel estimation

**Balanced contrastive learning** [73] (BCL) was proposed to solve long-tailed problems with improved supervised contrastive learning. BCL involves two key techniques: class averaging and class complement. BCL averages out the contributions of different classes in the denominator to ensure class equal distribution, meanwhile, it takes nonlinear mapping of the classifier parameters to form a learnable class center to ensure that every class has at least one sample in a mini-batch:

$$
\mathbb{E}_{\boldsymbol{x}' \sim \mathbb{P}(\cdot \mid Y = k)} \left[\kappa\left(\boldsymbol{z}_{\boldsymbol{x}}, \boldsymbol{z}_{\boldsymbol{x}'}\right)\right] \approx \frac{1}{|\mathcal{B}_k| + 1} \sum_{\boldsymbol{x}' \in \mathcal{B}_k \cup \{\boldsymbol{c}_k\}} \exp\left(\boldsymbol{z}_{\boldsymbol{x}} \cdot \boldsymbol{z}_{\boldsymbol{x}'}\right).
\tag{25}
$$

However, BCL ignores the problem of the original long-tailed distribution in the training dataset, necessitating a reweighting operation. Let $\boldsymbol{\pi}$ denote the class prior, we have:

$$
\widehat{\mathcal{L}}_{\mathrm{bcl}}(\boldsymbol{x}, y) = -\frac{1}{\pi_y} \log \frac{\frac{1}{|\mathcal{B}_y|} \sum_{\boldsymbol{x}' \in \mathcal{B}_y \cup \{\boldsymbol{c}_y\} \setminus \{\boldsymbol{x}\}} \exp\left(\boldsymbol{z}_{\boldsymbol{x}} \cdot \boldsymbol{z}_{\boldsymbol{x}'}\right)}{\sum_{k \in [C]} \frac{1}{|\mathcal{B}_k| + 1} \sum_{\boldsymbol{x}' \in \mathcal{B}_k \cup \{\boldsymbol{c}_k\}} \exp\left(\boldsymbol{z}_{\boldsymbol{x}} \cdot \boldsymbol{z}_{\boldsymbol{x}'}\right)}.
\tag{26}
$$

**Gaussian mixture likelihood loss** [57] (GML) initiates its approach from the concept of mutual information, employing the Gaussian kernel. GML integrates contrastive learning with logit adjustment to enhance its performance.

$$
\hat{\mathcal{L}}_{\mathrm{gml}}(\boldsymbol{x}, y) = -\log \frac{\frac{1}{|\mathcal{B}_y| + |\mathcal{Q}_y| - 1} \sum_{\boldsymbol{x}' \in \mathcal{B}_y \cup \mathcal{Q}_y \setminus \{\boldsymbol{x}\}} \exp\left(\boldsymbol{z}_{\boldsymbol{x}} \cdot \boldsymbol{z}_{\boldsymbol{x}'}\right) \pi_y}{\sum_{k \in [C]} \frac{1}{|\mathcal{B}_k| + |\mathcal{Q}_k|} \sum_{\boldsymbol{x}' \in \mathcal{B}_k \cup \mathcal{Q}_k} \exp\left(\boldsymbol{z}_{\boldsymbol{x}} \cdot \boldsymbol{z}_{\boldsymbol{x}'}\right) \pi_k}.
\tag{27}
$$

It proposes a class-wise queue $\mathcal{Q} = \cup_{k=1}^{C} \mathcal{Q}_k$ to ensure a balanced class occurrence in a mini-batch. However, it does not propose a unified framework, and the understanding is not deep enough.

**Some other methods:** In this section, we compare some other methods that use contrastive learning and analyze their mistakes.

**K-positive contrastive learning** [30] (KCL) is based on supervised contrastive learning, using K samples of the same class in the molecule to ensure balanced feature space. Putting in our unified framework, we can obtain the following:

$$\mathbb{E}_{\boldsymbol{x}'\sim\mathbb{P}(\cdot|Y=y)}\left[\kappa\left(\boldsymbol{z}_{\boldsymbol{x}},\boldsymbol{z}_{\boldsymbol{x}'}\right)\right] \approx \frac{1}{|K|}\sum_{\boldsymbol{x}'\in\mathcal{B}';\mathcal{B}'\subseteq\mathcal{B}_y,|\mathcal{B}'|=K}\exp\left(\boldsymbol{z}_{\boldsymbol{x}}\cdot\boldsymbol{z}_{\boldsymbol{x}'}\right),\tag{28}$$

$$\widehat{\mathcal{L}}_{\text{kcl}}^{\text{in}}(\boldsymbol{x},y) = -\log\frac{\frac{1}{|K|}\sum_{\boldsymbol{z}_{\boldsymbol{x}'}\in\mathcal{B}';\mathcal{B}'\subseteq\mathcal{B}_y,|\mathcal{B}'|=K}\exp\left(\boldsymbol{z}_{\boldsymbol{x}}\cdot\boldsymbol{z}_{\boldsymbol{x}'}\right)}{\sum_{\boldsymbol{z}_{\boldsymbol{x}'}\in\mathcal{B}}\exp\left(\boldsymbol{z}_{\boldsymbol{x}}\cdot\boldsymbol{z}_{\boldsymbol{x}'}\right)}$$

$$\overset{\text{Jensen}}{\leq} -\frac{1}{|K|}\sum_{\boldsymbol{z}_{\boldsymbol{x}'}\in\mathcal{B}';B'\subseteq\mathcal{B}_y,|\mathcal{B}'|=K}\log\frac{\exp\left(\boldsymbol{z}_{\boldsymbol{x}}\cdot\boldsymbol{z}_{\boldsymbol{x}'}\right)}{\sum_{\boldsymbol{z}_{\boldsymbol{x}'}\in\mathcal{B}}\exp\left(\boldsymbol{z}_{\boldsymbol{x}}\cdot\boldsymbol{z}_{\boldsymbol{x}'}\right)} = \widehat{\mathcal{L}}_{\text{kcl}}^{\text{out}}(\boldsymbol{x},y).\tag{29}$$

Compared with the above, it regrettably still uses the same unprocessed denominator of SCL and cannot ensure that each mini-batch contains an equal number of samples from each class, nor that each class contributes equally, rendering it suboptimal for long-tailed learning. In addition, Eq. (28) is equivalent to the resampling technique.

**Parametric contrastive learning** [15] (PaCo) introduces a set of parametric class-wise learnable centers and uses adjustable parameters $\alpha$ for loss with respect to them. Our framework is used to derive its original form. First, we can obtain:

$$\mathbb{E}_{\boldsymbol{x}'\sim\mathbb{P}(\cdot|Y=k)}\left[\kappa\left(\boldsymbol{z}_{\boldsymbol{x}},\boldsymbol{z}_{\boldsymbol{x}'}\right)\right] \approx \frac{\beta}{|\mathcal{B}_k|}\sum_{\boldsymbol{x}'\in\mathcal{B}_k}\exp\left(\boldsymbol{z}_{\boldsymbol{x}}\cdot\boldsymbol{z}_{\boldsymbol{x}'}\right) + (1-\beta)\exp\left(\boldsymbol{z}_{\boldsymbol{x}}\cdot\boldsymbol{c}_{\boldsymbol{k}}\right)\tag{30}$$

where $\beta$ is a fixed coefficient. Then, we can derive the PaCo loss as follows.

$$\widehat{\mathcal{L}}_{\text{paco}}(\boldsymbol{x},y) =$$

$$-\log\frac{\left(\frac{\beta}{|\mathcal{B}_y|}\sum_{\boldsymbol{x}'\in\mathcal{B}_y\setminus\{\boldsymbol{x}\}}\exp\left(\boldsymbol{z}_{\boldsymbol{x}}\cdot\boldsymbol{z}_{\boldsymbol{x}'}\right) + (1-\beta)\exp\left(\boldsymbol{z}_{\boldsymbol{x}}\cdot\boldsymbol{c}_{\boldsymbol{y}}\right)\right)\mathbb{P}(Y=y)}{\sum_{k\in[C]}\left(\frac{\beta}{|\mathcal{B}_k|}\sum_{\boldsymbol{x}'\in\mathcal{B}_k}\exp\left(\boldsymbol{z}_{\boldsymbol{x}}\cdot\boldsymbol{z}_{\boldsymbol{x}'}\right) + (1-\beta)\exp\left(\boldsymbol{z}_{\boldsymbol{x}}\cdot\boldsymbol{c}_{\boldsymbol{k}}\right)\right)\mathbb{P}(Y=k)}\tag{31}$$

$$\approx -\log\frac{\left(\alpha\sum_{\boldsymbol{x}'\in\mathcal{B}_y\setminus\{\boldsymbol{x}\}}\exp\left(\boldsymbol{z}_{\boldsymbol{x}}\cdot\boldsymbol{z}_{\boldsymbol{x}'}\right) + \exp\left(\boldsymbol{z}_{\boldsymbol{x}}\cdot\boldsymbol{c}_{\boldsymbol{y}}\right)\right)\mathbb{P}(Y=y)}{\sum_{k\in[C]}\left(\alpha\sum_{\boldsymbol{x}'\in\mathcal{B}_k}\exp\left(\boldsymbol{z}_{\boldsymbol{x}}\cdot\boldsymbol{z}_{\boldsymbol{x}'}\right) + \exp\left(\boldsymbol{z}_{\boldsymbol{x}}\cdot\boldsymbol{c}_{\boldsymbol{k}}\right)\right)\mathbb{P}(Y=k)}$$

where $\alpha = \frac{\beta\mathbb{P}(Y=y)}{(1-\beta)|\mathcal{B}_y|}$. PaCo explicitly uses the parametric class center to ensure balanced class occurrence in a mini-batch. However, It ignores the class-equal contribution in loss computation, which can still be suboptimal.

**Probabilistic contrastive learning** [20] (Proco) simply assumes that the normalized features in contrastive learning follows a mixture of von Mises-Fisher (vMF) distributions on a unit ball, its probability density function has the following form:

$$f_p\left(\boldsymbol{z};\boldsymbol{\mu}_y,\rho_y\right) = \frac{1}{C_p\left(\kappa_y\right)}e^{\rho\boldsymbol{\mu}^\top\boldsymbol{z}},$$

$$C_p(\rho) = \frac{(2\pi)^{p/2}I_{(p/2-1)}(\rho)}{\rho^{p/2-1}} \quad I_{(p/2-1)}(\boldsymbol{z}) = \sum_{k=0}^{\infty}\frac{1}{k!\Gamma(p/2-1+k+1)}\left(\frac{\boldsymbol{z}}{2}\right)^{2k+p/2-1}\tag{32}$$

where parameters $(\boldsymbol{\mu}_y,\rho_y)$ need to be estimated. The advantage is that Proco can estimate $(\boldsymbol{\mu}_y,\rho_y)$ using an online mini-batch, such that it can be derived as a closed form of expected contrastive loss. Despite the assumed vMF distribution, it still uses Gaussian kernel estimation:

$$\mathbb{E}_{\boldsymbol{x}'\sim\mathbb{P}(\cdot|Y=y)}\left[\kappa\left(\boldsymbol{z}_{\boldsymbol{x}},\boldsymbol{z}_{\boldsymbol{x}'}\right)\right] \approx \mathbb{E}_{\boldsymbol{z}_{\boldsymbol{x}'}\sim\widehat{\mathbb{P}}_{\text{vMF}}(\cdot|Y=y)}\left[\kappa\left(\boldsymbol{z}_{\boldsymbol{x}},\boldsymbol{z}_{\boldsymbol{x}'}\right)\right] = \frac{C_p(\lambda(\boldsymbol{z}_x,y))}{C_p\left(\rho_y\right)}\tag{33}$$

where $\lambda(\boldsymbol{z}_x,y)$ represents a fixed function related to $\boldsymbol{z}_x,\boldsymbol{\mu}_y,\rho_y$. Thus, the loss objective of Proco is:

$$\widehat{\mathcal{L}}_{\text{proco}}(\boldsymbol{x},y) = -\log\widehat{\mathbb{P}}_s(Y=y\mid\boldsymbol{z}) = -\log\frac{\frac{C_p(\lambda(\boldsymbol{z}_x,y))\cdot\pi_y}{C_p(\rho_y)}}{\sum_{k\in[C]}\frac{C_p(\lambda(\boldsymbol{z}_x,k))\cdot\pi_k}{C_p(\rho_k)}}.\tag{34}$$

However, the assumed distribution of Proco stills needs to be estimated $(\boldsymbol{\mu}_y,\rho_y)$ using EMA of different batches, which is essentially a similar approach to the momentum queue used in GML. There still exist problems of inconsistent distribution of $\boldsymbol{z}$ in different mini-batches, and the strong assumption about $\mathbb{P}(\boldsymbol{z}\mid Y=y)$ which may not follow the vMF distribution.

## B.2 Discussion of explicitly assigning a specified distribution

Previous work mainly focused on $\mathbb{P}(\boldsymbol{z} \mid Y = y)$ through modeling $\cos \theta_y = \frac{\boldsymbol{\mu}_y^\top \boldsymbol{z}}{\|\boldsymbol{\mu}_y\| \|\boldsymbol{z}\|}$.

**T-vMF** [34] models $\cos \theta_y$ as vMF distribution:

$$f\left(\cos \theta_y; \rho_y\right) = \frac{1}{C\left(\rho_y\right)} e^{\rho_y \cos \theta_y} = C'(\rho_y) e^{\rho_y \|\boldsymbol{z} - \boldsymbol{\mu}_y\|} = C'(\rho_y) s_e(\|\boldsymbol{z} - \boldsymbol{\mu}_y\|, \rho_y). \quad (35)$$

Due to the inherent properties of the exponential function, the posterior quickly converges to 0, despite a large $\|\boldsymbol{z} - \boldsymbol{\mu}_y\|$. Such a compact measuring function might hamper model training, since tailed samples hardly enjoy back-propagation due to vanishing gradient. To overcome this problem, T-vMF introduces a family of modeling methods as follows:

$$f_q\left(\cos \theta_y; \rho_y\right) = C'(\rho_y) \left[1 - (1 - q)\frac{1}{2} \rho_y \|\boldsymbol{z} - \boldsymbol{\mu}_y\|\right]^{\frac{1}{1-q}} = C'(\rho_y) s_q(\|\boldsymbol{z} - \boldsymbol{\mu}_y\|, \rho_y), \quad (36)$$

where $s_q(\|\boldsymbol{z} - \boldsymbol{\mu}_y\|, \rho_y) = \left[1 - (1 - q)\frac{1}{2} \rho_y \|\boldsymbol{z} - \boldsymbol{\mu}_y\|\right]^{\frac{1}{1-q}}$. Technically, T-vMF models $\widehat{\mathbb{P}}_t(Y = y \mid \boldsymbol{z})$ as follows:

$$\widehat{\mathbb{P}}_t(Y = y \mid \boldsymbol{z}) = \frac{e^{\varphi_{q,\rho} \langle \boldsymbol{z}, \boldsymbol{\mu}_y \rangle}}{\sum_{k \in [C]} e^{\varphi_{q,\rho} \langle \boldsymbol{z}, \boldsymbol{\mu}_k \rangle}}, \quad (37)$$

$$\varphi_{q,\rho} \langle \boldsymbol{z}, \boldsymbol{\mu}_y \rangle = 2\frac{s_q(\|\boldsymbol{z} - \boldsymbol{\mu}_y -\|, \rho) - s_q(2, \rho)}{s_q(0, \rho) - s_q(2, \rho)} - 1 \in [-1, 1]. \quad (38)$$

Thus, the loss objective of T-vMF is:

$$\widehat{\mathcal{L}}_{\text{T-vMF}}(\boldsymbol{x}, y) = -\log \widehat{\mathbb{P}}_s(Y = y \mid \boldsymbol{z}) = -\log \frac{\pi_y e^{\varphi_{q,\rho} \langle \boldsymbol{z}, \boldsymbol{\mu}_y \rangle}}{\sum_{k \in [C]} \pi_k e^{\varphi_{q,\rho} \langle \boldsymbol{z}, \boldsymbol{\mu}_k \rangle}}. \quad (39)$$

**WCDAS** [27] The accuracy of the posterior approximation is a crucial factor influencing the method's performance. Unlike t-vMF, which directly specifies the $\mathbb{P}(\boldsymbol{z} \mid Y = y)$ with fixed parameters, WCDAS seeks an optimal parametric probability density function of $\mathbb{P}(\boldsymbol{z} \mid Y = y)$.

Modeling $\cos \theta_y$ as the Wrapped Cauchy distribution with trainable parametric $\boldsymbol{\vartheta} = [\vartheta_1, \ldots, \vartheta_C]$, WCDAS models $\widehat{\mathbb{P}}_t(Y = y \mid \boldsymbol{z})$ as follows:

$$f\left(\cos \theta_y; \vartheta_y\right) = \frac{1 - \vartheta_y^2}{2\Pi(1 + \vartheta_y^2 - 2\vartheta_y \cos \theta_y)}, \quad (40)$$

$$\widehat{\mathbb{P}}_t(Y = y \mid \boldsymbol{z}) = \frac{e^{f(\cos \theta_y; \vartheta_y)}}{\sum_{k \in [C]} e^{f(\cos \theta_k; \vartheta_k)}}. \quad (41)$$

Thus, the loss objective of WCDAS is:

$$\widehat{\mathcal{L}}_{\text{WCDAS}}(\boldsymbol{x}, y) = -\log \widehat{\mathbb{P}}_s(Y = y \mid \boldsymbol{z}) = -\log \frac{\pi_y e^{f(\cos \theta_y; \vartheta_y)}}{\sum_{k \in [C]} \pi_k e^{f(\cos \theta_k; \vartheta_k)}}. \quad (42)$$

# C  Mathematical Notations

To ensure clarity and precision throughout this paper, we provide a comprehensive list and definitions of the key mathematical symbols and terms used in this section. Each symbol is defined with its specific meaning and context to ensure consistency and accuracy across the document.

# D  Illustration of The Proposed Algorithm

## D.1  Illustration of The Overall Proposed Algorithm

CCL consists of two parts: the classification part and the contrastive learning part. The classification part uses logit rectification of the classifier by class prior estimated with a dual-branch. For the contrastive learning part, the energy score is used to select reliable unlabeled data which are merged with labeled data for continuous contrastive loss to ensure calibration. Besides, information of labeled data and unlabeled are used in a decoupled manner while maintaining the constraints of aligning feature in the contrastive learning space, thereby forming a smoothed contrastive loss.

Table 7: List of common mathematical symbols used in this paper.

| Symbol | Definition |
|---|---|
| $\mathcal{X} \subset \mathbb{R}^d, \mathcal{Y} \subset [C]$ | Feature space and target space |
| $P_s, P_t$ | Joint distribution of training and test data in LTL, respectively |
| $P_l, P_u$ | Joint distribution of labeled and unlabeled data in LTSSL, respectively |
| $\{(\boldsymbol{x}_i, y_i)\}_{i=1}^N \sim P_s^N$ | Training set in LTL |
| $\{(\boldsymbol{x}_i^l, y_i^l)\}_{i=1}^N \sim P_l^N$ | Labeled training set in LTSSL |
| $\{\boldsymbol{x}_j^u\}_{j=1}^M \sim P_u^M$ | Unlabeled training set in LTSSL |
| $\{N_1, \ldots, N_C\}$ | Number of samples for each class in labeled data |
| $\{M_1, \ldots, M_C\}$ | Number of samples for each class in unlabeled data |
| $\boldsymbol{\pi}^l, \widehat{\boldsymbol{\pi}}^u$ | True class prior of labeled data and estimated one of unlabeled data, respectively |
| $X, Y, Z$ | Random variable of input, target, and latent feature space, respectively |
| $\mathbb{P}, \widehat{\mathbb{P}}$ | Probability density function and its variational approximation, respectively |
| $\mathbb{P}_s$ | Probability density function of training distribution in LTL |
| $\mathbb{P}_t$ | Probability density function of test distribution (with uniform label distribution) |
| $\mathbb{P}_l, \mathbb{P}_u$ | Probability density function of $L, U$, respectively |
| $\widehat{\mathbb{P}}^{\text{cls}}$ | Estimated posterior of unlabeled data with dual-branch |
| $\text{enc}(\cdot)$ | Encoder that maps $X$ to $Z$ |
| $\boldsymbol{\Theta}$ | Parameters of the encoder |
| $I(\cdot)$ | Mutual information of two random variables |
| $\kappa(\cdot, \cdot)$ | Similarity between two latent features |
| $\mathcal{M}, \mathcal{M}^E$ | Sample mask of confidence and energy score, respectively |
| $f_s(\cdot), f_b(\cdot)$ | Standard branch and balanced branch, respectively |
| $g(\cdot)$ | Projection head |
| $\mathcal{B}$ | Data mini-batch |
| $\mathcal{B}^l, \mathcal{B}^u$ | Mini-batch of labeled and unlabeled data, respectively |

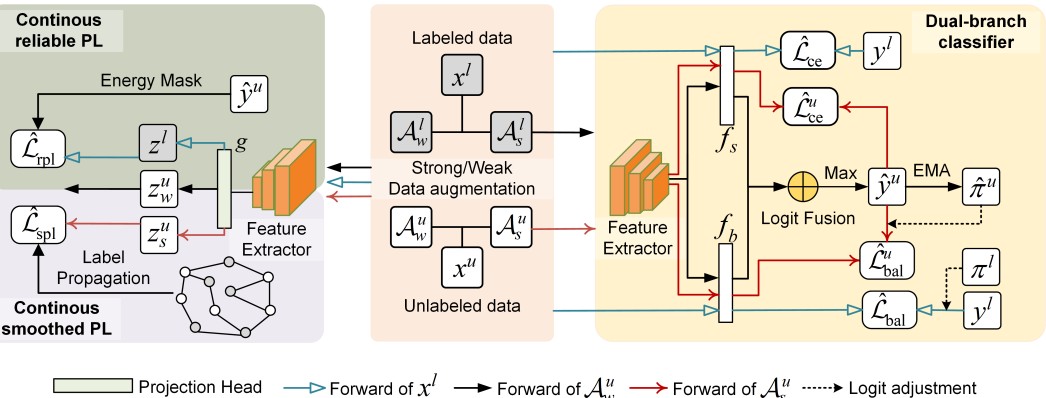

Figure 3: Illustration of the proposed framework.

---

**Algorithm 1** Continous Contrastive Learning (CCL)

---

**Input:** labeled dataset and unlabeled dataset, standard branch $f_s$ and balanced branch $f_b$, projection head $g$, class prior of labeled dataset $\pi_l$, estimated unlabeled dataset class distribution $\widehat{\pi}_u$, number of iterations in each epoch $T$, scaling parameter $\tau$.

**Require:** Weak augmentation $\mathcal{A}_w(\cdot)$, strong augmentation $\mathcal{A}_s(\cdot)$, loss weight coefficients $\lambda_1, \lambda_2$.

**for** $t = 1$ **to** $T$ **do**

    $\{(x_i^{(l)}, y_i^{(l)})\}_{i=1}^{|\mathcal{B}^l|} \leftarrow$ Sample a batch of labeled data

    $\{x_j^{(u)}\}_{j=1}^{|\mathcal{B}^u|} \leftarrow$ Sample a batch of unlabeled data

    # Balanced classifier training with estimated class prior

    Calculate pseudo label $\widehat{y} = \arg\max_{k \in [C]} \widehat{\mathbb{P}}^{\text{cls}}\left(Y = k \mid \mathcal{A}_w(\boldsymbol{x}^u)\right)$ via Eq. (15)

    Calculate classification loss $\widehat{\mathcal{L}}_{\text{cls}}$

    Update estimated class distribution $\widehat{\boldsymbol{\pi}}^u$ via EMA by energy score selection

    # Continuous contrastive loss with reliable pseudo-labels

    Merge reliable unlabeled data selected based on energy score with labeled data to construct $\mathcal{B}$

    Calculate loss $\widehat{\mathcal{L}}_{\text{rpl}}$ via Eq. (17) with $\mathcal{B}$

    # Continuous contrastive loss with smoothed pseudo-labels

    Calculate $\boldsymbol{G}$ using unlabeled data

    Compute posterior $\widehat{\mathbb{P}}\left(Y \mid \mathcal{A}_w(\boldsymbol{x}^u)\right)$ and $\widehat{\mathbb{P}}\left(Y \mid \mathcal{A}_s(\boldsymbol{x}^u)\right)$ using Eq. (21)[3]

    Calculate consistency regularization loss $\widehat{\mathcal{L}}_{\text{spl}}$ via Eq. (18)

    # Total Objective

    $\widehat{\mathcal{L}}_{\text{total}} = \lambda_1 \widehat{\mathcal{L}}_{\text{cls}} + (1 - \lambda_1)\widehat{\mathcal{L}}_{\text{rpl}} + \lambda_2 \widehat{\mathcal{L}}_{\text{spl}}$

    Update $f_s$ and $f_b$ and $g$ based on $\nabla \mathcal{L}_{\text{total}}$ using SGD

**end for**

---

## D.2 Illustration of Reliable Pseudo-labels and Smoothed Pseudo-labels

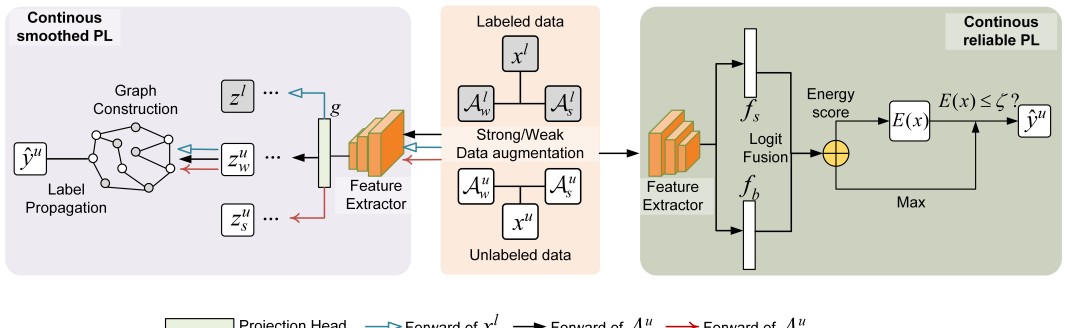

Figure 4: Illustration of reliable pseudo-labels and smoothed pseudo-labels in CCL. To generalize the framework in Section 2 to LTSSL, the main challenge is unknown $\mathbb{P}_u(Y = y \mid \boldsymbol{x}^u)$, where $\boldsymbol{x}^u$ denotes a sample in the unlabeled dataset. We first approximate $\mathbb{P}_u(Y = y \mid \boldsymbol{x}^u)$ using the output of the calibrated and integrated classifier and use energy score to filter out reliable unlabeled data, ensuring the model's calibration, which constitutes the reliable pseudo-labels subset. Furthermore, we can also estimate the unknown $\mathbb{P}_u(Y = y \mid \boldsymbol{x}^u)$ by leveraging the smoothness assumption. Specifically, we construct smoothed pseudo-labels by propagating labels from nearby samples using the Gaussian kernel density estimation.

## E Pseudo Code of The Proposed Algorithm

Algorithm 1 summarizes the whole framework of the proposed CCL, which is clearly divided into three components: balanced classifier, continuous contrastive learning with reliable and smoothed pseudo-labels, respectively.

## F  Experimental Setup

**Training datasets.** Our experimental analysis uses a variety of commonly adopted SSL datasets, including CIFAR10-LT [35], CIFAR100-LT [35], STL10-LT [13], and ImageNet-127 [22] in various ratios of class imbalance $\gamma$ and various ratios of the amount of labeled data $\eta$. To create imbalanced versions of these datasets, we consider the long-tailed imbalance where the frequency of data points decreases exponentially from the largest to the smallest class, that is, the number of samples in class $c$ is $N_c = N_1 \times \gamma^{-\frac{c-1}{C-1}}$ for labeled data and $M_c = M_1 \times \gamma^{-\frac{c-1}{C-1}}$ for unlabeled data. We use Cutout [19] and Randaugment [14] for strong augmentation on unlabeled data, and we use SimAugment [11] on labeled data for continuous contrastive loss with smoothed pseudo-labels. Like recent LTSSL works, we consider three class distribution patterns for unlabeled data, namely, `consistent`, `uniform`, and `reversed` settings.

- **CIFAR10-LT**: Following ACR [64], we conduct experiments with all comparison methods in settings where $N_1 = 500, M_1 = 4000$ and $N_1 = 1500, M_1 = 3000$. We adopt imbalance ratios of $\gamma_l = \gamma_u = 100$ and $\gamma_l = \gamma_u = 150$ for `consistent` settings, while for `uniform` and `reversed` settings, we use $\gamma_l = 100, \gamma_u = 1$ and $\gamma_l = 100, \gamma_u = 1/100$, respectively.
- **CIFAR100-LT**: Like CIFAR10-LT, we perform experiments in configurations where $N_1 = 50, M_1 = 400$ and $N_1 = 150, M_1 = 300$. For the `consistent` settings, we use imbalance ratios of $\gamma_l = \gamma_u = 10$ and $\gamma_l = \gamma_u = 20$. In contrast, for the `uniform` and `reversed` settings, we apply $\gamma_l = 10, \gamma_u = 1$ and $\gamma_l = 10, \gamma_u = 1/10$, respectively.
- **STL10-LT**: Given the absence of ground-truth labels for the unlabeled data of the STL10 dataset, we manage the experiments by adjusting the imbalance ratio of the labeled data. Following ACR, we consider the labeled imbalance ratio of $\gamma_l = 10$ or $\gamma_l = 20$.
- **ImageNet-127**: ImageNet127 was first introduced in an earlier research [29] and utilized in LTSSL by CReST. This dataset consolidates the 1000 classes [18] from ImageNet into 127 classes, grouping them according to the WordNet hierarchy. For ImageNet-127, we follow the original setting in CoSSL [22] ($\gamma_l = \gamma_u \approx 286$).

**Implementation details.** Our experimental configuration largely aligns with Fixmatch [56] and ACR [64]. Specifically, we apply the Wide ResNet-28-2 [70] architecture to implement our method on the CIFAR10-LT, CIFAR100-LT and STL10-LT datasets; and ResNet-50 on ImageNet-127. We adopt the common training paradigm that the network is trained with standard SGD [47, 49, 58] for 500 epochs, where each epoch consists of 500 mini-batches, and a batch size of 64 for both labeled and unlabeled data. We use a cosine learning rate decay [42] where the initial rate is 0.03, we set $\tau = 2.0$ for logit adjustment on all datasets, except for ImageNet-127, where $\tau = 0.1$. We set the temperature $T = 1$ and the threshold $\zeta = -8.75$ for the energy score following [69], and we set $\lambda_1 = 0.7, \lambda_2 = 1.0$ on CIFAR10/100-LT and $\lambda_1 = 0.7, \lambda_2 = 1.5$ on STL10-LT and ImageNet-127 datasets for the final loss. We set $\beta = 0.2$ in Eq. (21) for smoothed pseudo-labels loss. To show the effectiveness of our approach, we perform a comparative analysis with several existing LTSSL algorithms, including DARP [33], CReST [63], DASO [48], ABC [38], and TRAS [66]. We also consider the most popular LTSSL methods ACR [64] and CPE [43]. The performance evaluation of these methods is based on the top-1 accuracy metric on the test set. We present the mean and standard deviation of the results from three independent runs for each method. In addition, our method is implemented using the PyTorch library and experimented on an NVIDIA RTX A6000 (48 GB VRAM) with an Intel Platinum 8260 (CPU, 2.30GHz, 220 GB RAM).

## G  In-depth Analysis

### G.1  Sensitive analysis of hyperparameters

As outlined in figure 5a, CCL is relatively robust to the fluctuation of $\beta$ from 0.1 to 0.4. However, when $\beta$ is set to 0, the propagation within unlabeled data is ignored, resulting in a performance decrease of about $0.9\%$. Thus, the necessity of using Eq. (21) is verified. In addition, figures 5b and 5c both demonstrate that CCL is robust to loss weighting coefficients $\lambda_1$ and $\lambda_2$ within a certain range.

---

[3]The matrix inversion operation is implemented using `torch.inverse()`, which utilizes the fast singular value decomposition. The time complexity is $\mathcal{O}(|\mathcal{B}^u|^3)$.

However, it is worth noting that when $\lambda_1 = 1.0$, the proposed continuous reliable pseudo-labels loss is ignored, resulting in performance degradation.

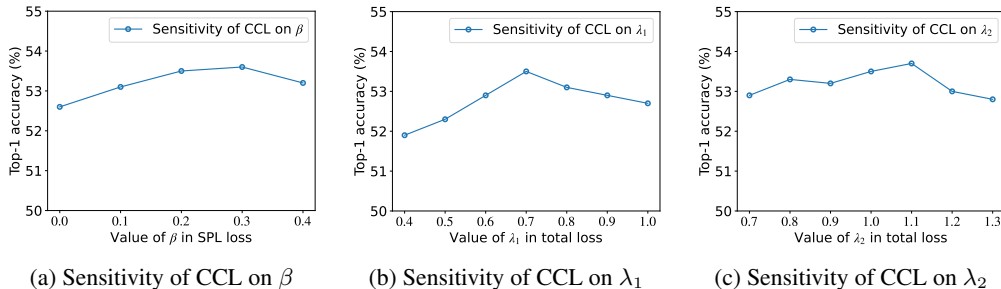

(a) Sensitivity of CCL on $\beta$     (b) Sensitivity of CCL on $\lambda_1$     (c) Sensitivity of CCL on $\lambda_2$

Figure 5: Sensitive analysis of hyperparameters under `consistent` setting of CIFAR100-LT.

## G.2 Time and Space Complexity Analyses

In this section, we conduct analyses of the time and space complexity of the proposed method. Denote feature space dimension $D$, batch size $B$ and the number of classes $C$, the time and space complexity of CCL can be seen in Table 8 and analysis details as follows.

For time complexity, calculating $\mathcal{L}_{rpl}$ requires two main parts, calculating kernel similarities by multiplying two matrix of size $B \times D$ and $D \times B$ with complexity $\mathcal{O}(B^2 D)$, and calculating $\mathbb{E}_{\boldsymbol{x}' \sim \mathbb{P}(\cdot | Y = y)}[\kappa(\boldsymbol{z}_{\boldsymbol{x}}, \boldsymbol{z}_{\boldsymbol{x}'})]$ by multiplying two matrix of size $B \times B$ and $B \times C$ with complexity $\mathcal{O}(B^2 C)$. Calculating $\mathcal{L}_{spl}$ requires a further calculating part compared to $\mathcal{L}_{rpl}$: inverse matrix $\boldsymbol{I} - \beta \boldsymbol{G}$ in Eq.(21) with complexity $\mathcal{O}(B^3)$ by utilizing the fast singular value decomposition.

For space complexity, calculating two losses requires two additional storage spaces, first sample pairwise kernel similarity requires space with complexity $\mathcal{O}(B^2)$, and $\widehat{\mathbb{P}}(Y | \boldsymbol{X}^u)$ requires space with complexity $\mathcal{O}(BC)$.

Generally, given $B = 64$, $C = 100$ and $D = 256$. Compared to the scale of model parameters, CCL adds negligible overhead relative to the neural network's computational cost when computing loss. We further report the averaged mini-batch training time with a single 3090 GPU and the GPU memory usage in Table 9 and Table 10. As seen from these tables, the training time and space consumptions of CCL are comparable to the existing state-of-the-art method ACR when CCL applies additional data augmentations to labeled data for representation learning.

Table 8: Time and space complexity of two continuous contrastive loss of CCL.

| CCL loss | Time complexity | Space complexity |
|---|---|---|
| $\mathcal{L}_{\text{rpl}}$ | $\mathcal{O}(B^2 D + B^2 C)$ | $\mathcal{O}(B^2 + BC)$ |
| $\mathcal{L}_{\text{spl}}$ | $\mathcal{O}(B^2 D + B^2 C + B^3)$ | $\mathcal{O}(B^2 + BC)$ |

Table 9: Average batch time of each algorithm.

| Algorithm | CIFAR-10 | CIFAR-100 | STL-10 |
|---|---|---|---|
| ACR | 0.073 sec/iter | 0.083 sec/iter | 0.114 sec/iter |
| CCL | 0.102 sec/iter | 0.111 sec/iter | 0.143 sec/iter |

## G.3 Confusion matrix

Figure 6 presents the confusion matrix on the test set generated by CCL and ACR, which is calculated on the CIFAR10-LT dataset under $\gamma_l = \gamma_u = 100$ and $\gamma_l = \gamma_u = 150$ settings. As we can see in the top row of the figure, ACR often misclassifies the minority class "7" and "8" into the majority class "4" and "0", respectively. In comparison, CCL effectively mitigates this misclassification phenomenon by achieving an average improvement of 7.5%. Similarly in the

Table 10: GPU memory usage of each algorithm.

| Algorithm | CIFAR-10 | CIFAR-100 | STL-10 |
|-----------|----------|-----------|--------|
| ACR | 2054M | 2057M | 2236M |
| CCL | 2230M | 2232M | 2642M |

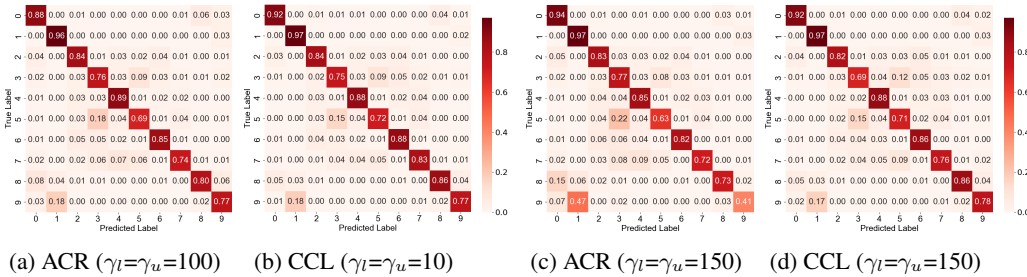

(a) ACR ($\gamma_l=\gamma_u=100$)  (b) CCL ($\gamma_l=\gamma_u=10$)  (c) ACR ($\gamma_l=\gamma_u=150$)  (d) CCL ($\gamma_l=\gamma_u=150$)

Figure 6: Confusion matrices of the predictions on the test set of CIFAR10-LT.

second row, CCL achieved an extraordinarily high accuracy of 78% for class "9", which shows a significant gain of 37% compared to ACR. CCL also achieves higher overall accuracy.

### G.4 Precision and recall

To conduct a more in-depth analysis of the effectiveness of pseudo-labels generated by the proposed dual-branch fusion approach, we calculated the precision and recall of the pseudo-labels assigned to unlabeled data by ACR and CCL on CIFAR10-LT and CIFAR100-LT datasets. Specifically, we use $\gamma_l = \gamma_u = 100$ and $\gamma_l = \gamma_u = 150$ settings on CIFAR10-LT dataset and all three `consistent`, `uniform`, `reversed` settings on CIFAR100-LT dataset and we grouped the results of CIFAR100 into 10 categories, each category containing 10 classes, since CIFAR100 comprises 100 classes. As can be seen in figure 8, CCL achieves significantly improved precision of pseudo-labels for tailed classes "9" and "10" on CIFAR10 dataset, while also achieving better recall for head classes. Similarly in Figure 7, CCL achieves overall better precision and recall compared to ACR regardless of the distribution mismatch scenario. It clearly shows that the pseudo-labels generated by CCL are more capable of alleviating the confirmation bias of tailed classes without sacrificing the performance of head classes.

### G.5 Visualization

Furthermore, we employ the t-distributed stochastic neighbor embedding (t-SNE) [61] to visualize the representations learned by the CCL method and contrast them with those from the previous ACR method. The comparative results on the test set, under `consistent` settings, are depicted in Figure 9. The figure demonstrates that the representations derived from CCL provide more distinct classification boundaries.

## H  Limitation

Our paper examines existing long-tailed learning methods through the lens of information theoretical view, proposing a unified framework. However, we have not established theoretical proof for the convergence of features within this framework. In the future, we intend to provide further theoretical analysis from the perspective of neural collapse.

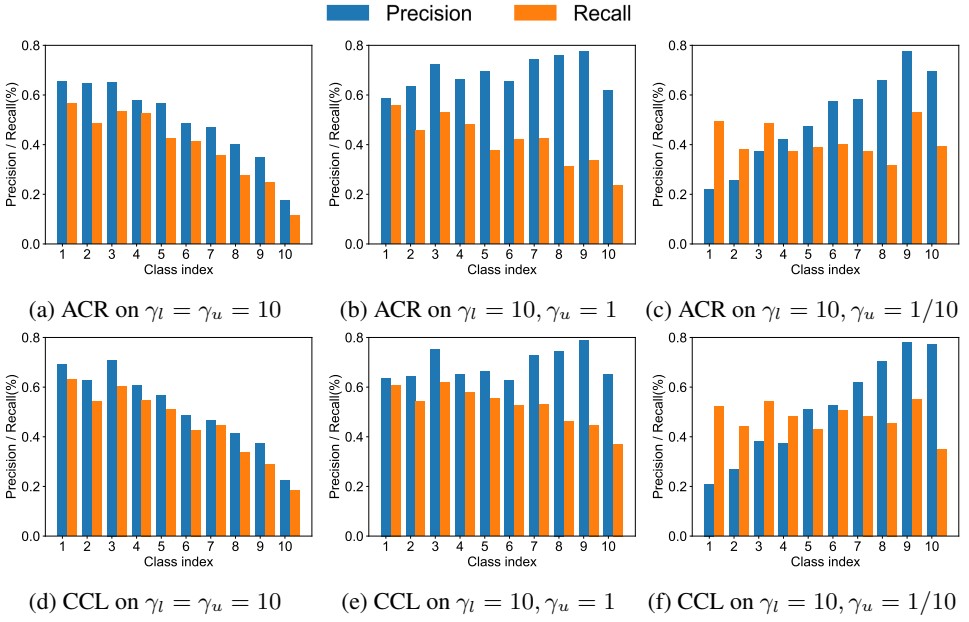

Figure 7: The precision and recall of pseudo-labels for ACR and CCL on CIFAR100-LT dataset in `consistent`, `uniform`, `reversed` settings.

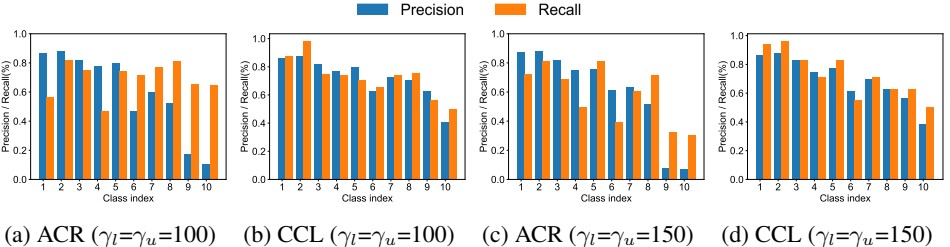

Figure 8: The precision and recall of pseudo-labels for ACR and CCL on CIFAR10-LT dataset in `consistent` settings.

# I   Broader Impact

The positive impacts of this work are two-fold: 1) It enhances the fairness of the classifier in semi-supervised learning, preventing potential biases in deep models, such as an unfair AI primarily serving the majority, which could lead to discrimination based on gender, race, or religion; 2) It enables the easy collection of larger image datasets without the need for mandatory class-balancing preprocessing. For example, in training classifiers for real-world natural image scenes using the proposed method, we do not need to consider whether the distribution of unlabeled data matches that of the labeled data or if every class in the labeled data has an equal number of samples. However, negative effects might occur if the proposed long-tailed semi-supervised classification technique is misused. In the wrong hands, this approach could be exploited for unethical purposes, such as targeting or identifying minority groups for detrimental reasons.

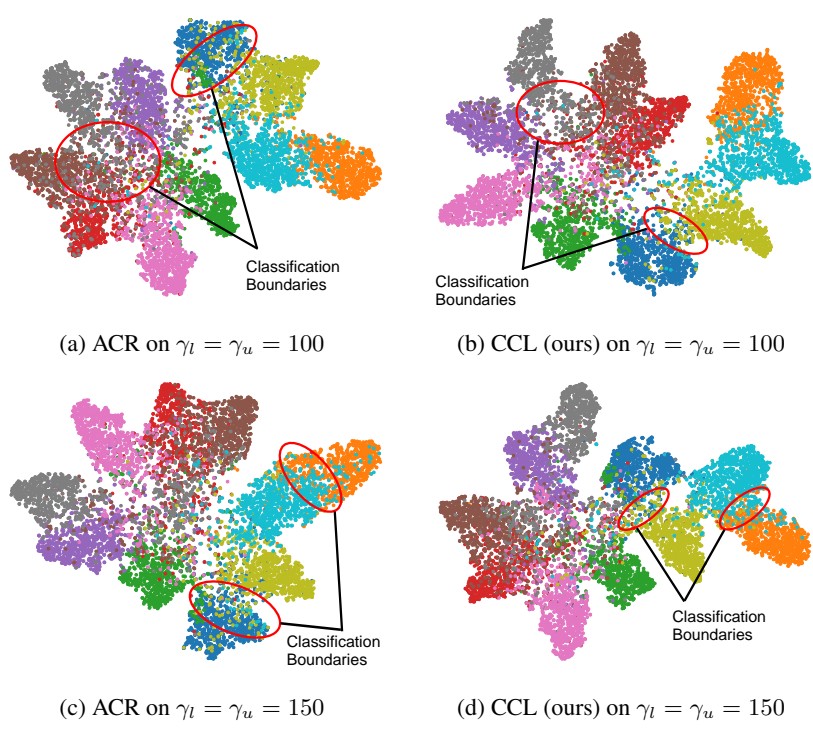

(a) ACR on $\gamma_l = \gamma_u = 100$

(b) CCL (ours) on $\gamma_l = \gamma_u = 100$

(c) ACR on $\gamma_l = \gamma_u = 150$

(d) CCL (ours) on $\gamma_l = \gamma_u = 150$

Figure 9: The t-SNE visualization of the test set for ACR and CCL on CIFAR-10-LT dataset in `consistent` settings.

