# OpenReview forum: "Continuous Contrastive Learning for Long-Tailed Semi-Supervised Recognition"
_NeurIPS.cc/2024/Conference — NeurIPS 2024 poster_

### Official Review · Reviewer_JsM2 · 2024-07-02

**Soundness:** 3
**Presentation:** 3
**Contribution:** 3
**Rating:** 7
**Confidence:** 5

**Summary:**

The paper proposes a novel method to address the long-tail problem in semi-supervised learning by leveraging continuous contrastive learning on both labeled and unlabeled samples to improve model performance, particularly for minority classes.

**Strengths:**

- Novelty: The paper introduces a novel approach to integrate contrastive learning with unlabeled samples into SSL frameworks, which is relatively unexplored in existing literature.
- Technical Soundness: The theoretical formulation of learning good representations and the unified framework of many existing long-tail learning methods are well-explained and logically sound.
- Experiments: The experiments are comprehensive, covering various imbalanced ratios and labeled ratios on CIFAR-10-LT and CIFAR-100-LT datasets. The results are statistically significant and show clear improvements over baseline methods.
- Clarity: The paper is well-structured and clearly written. The introduction, methodology, and experimental sections are easy to follow, and the figures and tables effectively illustrate the results.

**Weaknesses:**

- In line 189, it is unclear why selecting pseudo-labels using energy score can ensure better model calibration.

**Questions:**

A minor suggestion: the paper can include some demonstrations of unlabeled images with their associated continuous pseudo-labels, including both reliable and smoothed versions. This can help readers identify cases in which the model tends to make mistakes.

**Limitations:**

The authors have adequately addressed the limitations and potential negative societal impact of their work.

---

> ### Author Rebuttal · Authors · 2024-08-05
>
> Dear Reviewer JsM2,
>
> We sincerely appreciate your thoughtful feedback. We will address each of your concerns in detail.
>
> **Weakness #1: Why selecting pseudo-labels using energy score can ensure better model calibration**
>
> Since the confirmation bias is induced by **self-training**, using **confidence selection** may result in overconfident but wrong pseudo-labels and hurt the calibration[1,2].
>
> In contrast, the **energy score leverages the probability density of the predictions**, exhibiting reduced vulnerability to overconfidence [3]. Thus, the energy function can alleviate confirmation bias caused by overconfidence, thereby obtaining a reliable data subset on which the model is calibrated.
>
> We also compare the calibration metric ECE value [4] between confidence threshold and energy score in Figure 1 of the paper, and the results confirm our idea.
>
> **Weakness #2: The suggestion to demonstrate the reliable and smoothed pseudo labels of the unlabeled data for better understanding**
>
> **We demonstrate the generation process of reliable pseudo-label and smoothed pseudo-label in the attached PDF**.
>
> To generalize the proposed supervised learning framework to LTSSL, the main challenge is unknown $\mathbb{P}_{u}(Y = y \mid \boldsymbol{x}^u)$, where $\boldsymbol{x}^u$ denote sample in unlabeled dataset.
>
> To address this issue, we first approximate $\mathbb{P}_{u}(Y = y \mid \boldsymbol{x}^u)$ using **the output of the calibrated and integrated classifier** as expressed in Eq. (15). Additionally, we use **energy score** to filter out reliable unlabeled data, ensuring the model's calibration on this data subset, which constitutes the reliable pseudo-labels subset.
>
> Furthermore, we can also estimate the unknown $\mathbb{P}_{u}(Y = y \mid \boldsymbol{x}^u)$ by leveraging the **smoothness assumption**. Specifically, we construct smoothed pseudo-labels by propagating labels from nearby samples using **Gaussian kernel density estimation**.
>
> We will include more detailed textual and graphical demonstrations in our next version of the paper.
>
> ###### **Reference**
>
> [1] Mishra, S., Murugesan, B., Ayed, I. B., Pedersoli, M., & Dolz, J. (2024). Do not trust what you trust: Miscalibration in Semi-supervised Learning. *arXiv preprint arXiv:2403.15567*.
>
> [2] Loh, C., Dangovski, R., Sudalairaj, S., Han, S., Han, L., Karlinsky, L., ... & Srivastava, A. (2022). On the Importance of Calibration in Semi-supervised Learning. *arXiv preprint arXiv:2210.04783*.
>
> [3] Liu, W., Wang, X., Owens, J., & Li, Y. (2020). Energy-based out-of-distribution detection. *Advances in neural information processing systems*, *33*, 21464-21475.
>
> [4] Guo, C., Pleiss, G., Sun, Y., & Weinberger, K. Q. (2017, July). On calibration of modern neural networks. In *International conference on machine learning* (pp. 1321-1330). PMLR.

---

> > ### Comment · Reviewer_JsM2 · 2024-08-11
> > **Keep the original ratings**
> >
> > Thanks for your response. I would like to keep the original ratings.

---

### Official Review · Reviewer_WTLw · 2024-07-04

**Soundness:** 3
**Presentation:** 2
**Contribution:** 3
**Rating:** 6
**Confidence:** 2

**Summary:**

The author proposes a probabilistic framework that unifies many recent proposals in long-tail learning. Specifically, for long-tailed semi-supervised learning,  a continuous contrastive learning method based on reliable and smoothed pseudo-labels to address confirmation bias and improve the quality of learned representations is proposed. The experiments demonstrate their effectiveness.

**Strengths:**

1) The author notices an important problem that existing approaches often assume the equivalence of the unlabeled data distribution with the labeled data, which is more practical.

2) The proposed probabilistic framework which unifies many recent proposals in long-tail learning is interesting.

3) The paper is well-written and easy to follow.

**Weaknesses:**

1) The author should discuss explicitly how the proposed method solves the unlabelled data  for diverse label distributions of unlabeled data and why other methods do not.

2) How to interpret " Furthermore, these methods primarily focus on correcting model outputs without delving into the role of representation learning in improving performance."  line 33-34.

3) The author may give an analysis of the relationship between  reliable pseudo-labels and smoothed pseudo-labels and how they play a different role in semi-supervised learning.

4) The time and space complexity should be analyzed in experiments between the proposed method and other methods.

**Questions:**

see weakness

**Limitations:**

yes

---

> ### Author Rebuttal · Authors · 2024-08-05
>
> Dear Reviewer WTLw,
>
> We sincerely appreciate the reviewer for thoughtful feedback. We address your concerns one by one.
>
> **Weakness #1: How the proposed method solves the unlabelled data for diverse label distributions of unlabeled data**
>
> Having an accurate estimation of unlabeled data prior $\widehat{\boldsymbol{\pi}}_u$ helps solve LTSSL like supervised LTL, which can simply achieve balanced training by logit adjustment. We propose to use EMA estimation of dual-branches by selecting data subset from mini-batch via **energy score**.
>
> Accurate estimation of $\widehat{\boldsymbol{\pi}}_u$ requires **calibrated model** [1]. Methods like ACR and CPE, using confidence-based sample selection, producing overconfident but incorrect pseudo-labels [2]. In contrast, **energy score leverages prediction probability density**, mitigating the overconfidence problem [3]. Thus, energy selection is used for a reliable unlabeled data subset, on which model is calibrated, enabling accurate estimation of $\widehat{\boldsymbol{\pi}}_u$.
>
> In Figure 1 of our paper, we empirically compared the estimation error of confidence and energy score.
>
> **Weakness #2: How to interpret "Furthermore, these methods primarily focus on correcting model outputs without delving into the role of representation learning in improving performance"**
>
> Recent LTSSL methods focus on adjusting the classifier output to overcome the long-tail effect. Methods like **DASO**, **ACR**, and **CPE** improve pseudo-label quality through techniques such as classifier output ensembling or logits scaling, ignoring the representation learning.
>
> In contrast, our method delves into the long-tail effect in **representation learning**. CCL introduces balanced FixMatch for **classifier training** and enhances representation learning with **reliable and smoothed pseudo-labels**, as described in Eq. (17) and Eq. (18).
>
> **Weakness #3: Relationship between reliable pseudo-labels and smoothed pseudo-labels**
>
> The main challenge in generalizing our supervised learning framework to LTSSL is the unknown of $\mathbb{P}_{u}(Y = y \mid \boldsymbol{x}^u)$ for unlabeled samples.
>
> We first approximate $\mathbb{P}_{u}(Y = y \mid \boldsymbol{x}^u)$ using integrated classifier output in Eq. (15). We use energy score to filter out reliable unlabeled samples and their pseudo-labels, ensuring that model is calibrated on this subset. Learning from **reliable pseudo-labels** enhances feature discrimination.
>
> However, self-training can introduce confirmation bias and negatively impact the representation learning. Thus we propose **smoothed pseudo-labels** for $\boldsymbol{x}^u$ by propagating labels from nearby samples, leveraging the smoothness assumption in the feature space without classifier information. The purpose of $\mathcal{L}_{\mathrm{spl}}$ is to align the smoothed pseudo-labels of $\mathcal{A}\_{w}(\boldsymbol{x}^u)$ and $\mathcal{A}\_{s}(\boldsymbol{x}^u)$, serving as "consistency regularization" in SSL.
>
> **Weakness #4: The time and space complexity**
>
> Denote feature space dimension $D$, batch size $B$ and number of classes $C$, the time and space complexity of CCL are as follows.
>
> ##### Time Complexity
>
> For $\widehat{\mathcal{L}}\_{\mathrm{rpl}}$:
>
> 1. Calculating kernel similarities: $\mathbb{R}^{B \times D} \times \mathbb{R}^{D \times B}$, with complexity $\mathcal{O}(B^{2}D)$.
> 2. Calculating $\mathbb{E}\_{\boldsymbol{x}^{\prime} \sim \mathbb{P}(\cdot| Y=y)}\left[\kappa\left(\boldsymbol{z_x}, \boldsymbol{z_{x^{\prime}}}\right)\right]$ using Gaussian kernels: $\mathbb{R}^{B \times B} \times \mathbb{R}^{B \times C}$, with complexity $\mathcal{O}(B^{2}C)$.
>
> Total: $\mathcal{O}(B^{2}D+ B^{2}C)$.
>
> For $\widehat{\mathcal{L}}\_{\mathrm{spl}}$:
>
> 1. Calculating kernel similarities: $\mathbb{R}^{B \times D} \times \mathbb{R}^{D \times B}$, with complexity $\mathcal{O}(B^2 D)$.
> 2. Calculating $\widehat{\mathbb{P}}(Y \left.=y \mid \boldsymbol{x}^u ; \mathcal{B}^l\right)$ based on Eq. (19): $\mathbb{R}^{B \times B} \times \mathbb{R}^{B \times C}$, with complexity $\mathcal{O}(B^2 C)$.
> 3. Inverse matrix $(\boldsymbol{I} - \beta \boldsymbol{G})^{-1}$ in Eq. (21): $\mathcal{O}(B^3)$.
>
> Total: $\mathcal{O}(B^2 D + B^2 C + B^3)$.
>
> ##### Space Complexity
>
> For $\widehat{\mathcal{L}}\_{\mathrm{rpl}}$:
>
> 1. Pairwise kernel similarity : $\mathcal{O}(B^2)$.
> 2. Calculating $\mathbb{E}\_{\boldsymbol{x}^{\prime} \sim \mathbb{P}(\cdot| Y=y)}\left[\kappa\left(\boldsymbol{z_x}, \boldsymbol{z_{x^{\prime}}}\right)\right]$ using Gaussian kernels: $\mathcal{O}(BC)$.
>
> Total: $\mathcal{O}(B^2 + BC)$.
>
> For $\widehat{\mathcal{L}}\_{\mathrm{spl}}$:
>
> 1. Computing $\widehat{\mathbb{P}}(Y = y \mid \boldsymbol{x}^u ; \mathcal{B}^l)$: $\mathcal{O}(BC)$.
> 2. Pairwise kernel similarity: $\mathcal{O}(B^2)$.
>
> Total: $\mathcal{O}(B^2 + BC)$.
>
> In Table 2 and Table 3 of the global response (PDF) of our rebuttal, we empirically report the **averaged minibatch training time** with a single 3090 GPU and the **GPU memory usage**.
>
> Generally, $B=64$, $D=256$, and $C=100$. Compared to the \# of model parameters, CCL adds negligible overhead relative to the neural network's computational cost. **In Table 2 of our global response (PDF), we should note that the training time of CCL is comparable to existing state-of-the-art method ACR.**
>
> Thank you for the valuable feedback again. We will address all the issues mentioned in the next version of the paper.
>
> ###### **Reference**
>
> [1] Garg, S., Wu, Y., Balakrishnan, S., & Lipton, Z. (2020). A unified view of label shift estimation. *Advances in Neural Information Processing Systems*, *33*, 3290-3300.
>
> [2] Loh, C., Dangovski, R., Sudalairaj, S., Han, S., Han, L., Karlinsky, L., ... & Srivastava, A. (2022). On the Importance of Calibration in Semi-supervised Learning. *arXiv preprint arXiv:2210.04783*.
>
> [3] Liu, W., Wang, X., Owens, J., & Li, Y. (2020). Energy-based out-of-distribution detection. *Advances in neural information processing systems*, *33*, 21464-21475.

---

### Official Review · Reviewer_biXa · 2024-07-12

**Soundness:** 3
**Presentation:** 3
**Contribution:** 3
**Rating:** 7
**Confidence:** 5

**Summary:**

This paper proposes a novel contrastive learning method for long-tailed semi-supervised learning (LTSSL). The method is motivated by variational information bottleneck for learning good representations and extends to unlabeled data using continuous pseudo-labels. This paper showcases strong empirical results on multiple LTSSL datasets and label distributions.

**Strengths:**

1.The proposed method is well-motivated. Previous works in LTSSL generally make contributions in learning balanced classifiers, while this paper examines representation learning.
2.This paper summarizes many existing methods for long-tail learning in a unified framework. It is interesting to show that the equivalence between supervised contrastive learning and Gaussian kernel estimation.
3.This paper proposes a new LTSSL method by optimizing the contrastive loss using two types of continuous pseudo-labels. The proposed method is novel in LTSSL.
4.Extensive experiments showcase that the proposed method achieves best results on all datasets. The paper also provides extensive in-depth studies to understand the effectiveness of the method.

**Weaknesses:**

1.In Table 1, the paper states that long-tail learning methods need to avoid none class samples in each min-batch during training. However, it is unclear how this issue is tackled in the proposed method CCL.
2.The training time of the proposed method is not reported. It is suggested to compare the training time with ACR.

**Questions:**

Please see Weaknesses.

**Limitations:**

The authors have adequately addressed the limitations and potential negative societal impact of their work.

---

> ### Author Rebuttal · Authors · 2024-08-05
>
> Dear Reviewer biXa,
>
> We sincerely appreciate the reviewer's thoughtful feedback. We will address your concerns individually.
>
> **Weakness #1: How does the proposed method address the issue of having mini-batches with no samples from certain classes during training**
>
> For our proposed method CCL, to avoid none class samples in each min-batch during training, we maintain a **learnable class prototype** for each class to facilitate the contrastive learning and address the aforementioned issue. The design of the learnable class prototypes is achieved through a nonlinear mapping of the classifier's weights.
>
> **Weakness #2: The comparison of training time overhead between the proposed method and ACR.**
>
> Thank you for the suggestion for considering the time overhead of the proposed method! We evaluated the average time taken to process a minibatch with a single 3090 GPU.
>
> ###### Table 1: Average batch time of each algorithm
>
> | Algorithm | CIFAR-10       | CIFAR-100      | STL-10         |
> | --------- | -------------- | -------------- | -------------- |
> | ACR       | 0.073 sec/iter | 0.083 sec/iter | 0.114 sec/iter |
> | CCL       | 0.102 sec/iter | 0.121 sec/iter | 0.153 sec/iter |
>
> Since CCL additionally considers representation learning compared to ACR, it applies corresponding data augmentations to labeled data as well, resulting in an overhead approximately 1.5 times that of ACR. However, as seen from the Table 1, the execution speed is nearly identical.

---

### Official Review · Reviewer_1xLE · 2024-07-24

**Soundness:** 4
**Presentation:** 3
**Contribution:** 3
**Rating:** 7
**Confidence:** 4

**Summary:**

This paper tackles the long-tailed semi-supervised learning problem. It first reviews recent works with a novel probabilistic framework. Based on this, it proposes a continuous contrastive learning method, CCL, to extend the framework to unlabeled data with pseudo-labels. Experiments show that it outperforms all recent works on the popular long-tailed semi-supervised learning benchmarks.

**Strengths:**

Overall, this paper is good.

1. With a framework from information theory perspective, it unified different approaches on the long-tailed learning topic.

2. It systematically studied the key ideas to tackle the long-tailed learning and semi-supervised learning tasks.

3. By proposing CCL, it unifies different ideas from long-tailed learning and semi-supervised learning, and the result is an effective approach to solve the LTSSL problem.

**Weaknesses:**

I only have one question:

In Eq. 22, it mentioned L_{cls}, which is also mentioned in Line 171. Is that the same as P^{cls} in Eq. 15?

**Questions:**

See weaknesses

In Eq. 22, it mentioned L_{cls}, which is also mentioned in Line 171. Is that the same as P^{cls} in Eq. 15?

**Limitations:**

Limitations and potential impacts are addressed in Appendix,

---

> ### Author Rebuttal · Authors · 2024-08-04
>
> Dear Reviewer 1xLE,
>
> We sincerely appreciate your thoughtful feedback. We will respond to each of your concerns individually.
>
> **Weakness #1: Misunderstanding between $\widehat{\mathcal{L}}_{\mathrm{cls}}$ in Eq. (22) and $\widehat{\mathbb{P}}^{\mathrm{cls}}\left(Y=y \mid \boldsymbol{x}^u\right)$ in Eq. (15)**
>
> To clarify, $\widehat{\mathcal{L}}_{\mathrm{cls}}$ represents **the loss function** for the classification part of the dual-branches $f_s(\cdot)$ and $f_b(\cdot)$.
>
> However, $\widehat{\mathbb{P}}^{\mathrm{cls}}\left(Y=y \mid \boldsymbol{x}^u\right)$ denotes **the pseudo-labels** for training dual-branches $f_s(\cdot)$ and $f_b(\cdot)$, where we propose to fuse the predictions of balanced and standard branches by Eq. (15) to obtain $\widehat{\mathbb{P}}^{\mathrm{cls}}\left(Y=y \mid \boldsymbol{x}^u\right)$.

---

### Author Rebuttal · Authors · 2024-08-05

We extend our gratitude to the reviewers for their constructive and valuable feedback.

We are encouraged by the reviewers' acknowledgment of the novelty of our continuous contrastive learning framework (R2, R4), its significant contribution (R1, R2, R3, R4), and its soundness (R1, R2, R3, R4). Reviewers also consider the theoretical perspective to be interesting (R2, R3) and robust (R4). In addition, the reviewers recognize the practical value of our framework under LTSSL (R3, R4), noting that our approach is effective, with empirical results that outperform existing methods (R1, R2, R3, R4). We are pleased with comments regarding the good presentation (R1, R4), the quality of writing (R3), and the clear motivation and structure (R2) of our work.

(We refer to **Reviewer 1xLE** as R1, **Reviewer biXa** as R2, **Reviewer WTLw** as R3, and **Reviewer JsM2** as R4.)

We have taken the reviewers' comments into consideration and have responded to each of them individually. After carefully reviewing the comments, we have identified two common issues among the reviewers' feedback. To address these concerns, we have included empirical results and illustrations, which can be found in the **attached PDF**.

- **Time and space complexity analysis**

  We provide a detailed analysis of the time and space complexity results for CCL and the existing state-of-the-art method ACR. This includes the average minibatch processing time with a single 3090 GPU and the GPU memory usage during training.

- **Role and relationship of the reliable and smoothed pseudo-labels**

  We illustrate the generation of reliable and smoothed pseudo-labels in the PDF to facilitate a clearer understanding of the roles and relationships of these two types of pseudo-labels.

We will address and resolve these issues in the next version of the paper.

Thank you for taking the time to review our work. We appreciate your efforts and feedback.

---

### Decision · Program_Chairs · 2024-09-25

**Decision:**

Accept (poster)

**Comment:**

The paper initially presents a unified probabilistic framework for long-tail learning and further extends it to a semi-supervised setting through a novel continuous contrastive learning approach. The results demonstrate significant improvements across all benchmarks. The rebuttal effectively addresses the reviewers' concerns, leading to unanimous positive feedback. After reviewing the comments and feedback, the AC  recommends acceptance.